# Testicular toxicity following chronic codeine administration is via oxidative DNA damage and up-regulation of NO/TNF-α and caspase 3 activities

Roland Akhigbe, Ayodeji Ajayi◉*

Department of Physiology, College of Medicine, Ladoke Akintola University of Technology, Ogbomoso, Oyo, Nigeria

* jy_ayodeji@yahoo.com

## Abstract

### Background

Codeine, a 3-methylmorphine, and other related opioids have been implicated in androgen suppression, although the associated mechanisms remain unclear.

### Aim

Therefore, the objective of the current study was to elucidate the in vivo molecular mechanisms underlying codeine-induced androgen suppression.

### Methods

This study made use of twenty-one healthy male rabbits, distributed into three groups randomly, control and codeine-treated groups. The control had 1ml of normal saline daily p.o. The codeine-treated groups received either 4mg/kg b.w of codeine or 10mg/kg b.w of codeine p.o. for six weeks. Reproductive hormonal profile, testicular weight, testicular enzymes, oxidative and inflammatory parameters, testicular DNA fragmentation, histological examination and apoptosis marker were evaluated to examine the effects of codeine use.

### Key findings

Oral administration of codeine resulted in testicular atrophy and alterations in testicular histomorphology, elevated testicular enzymes, and suppression of circulatory and intra-testicular testosterone. These changes were associated with a marked rise in oxidative markers and decline in the activities of testicular enzymatic antioxidants, as well as oxidative DNA damage, inflammatory response, testicular DNA fragmentation, and caspase-dependent apoptosis ($p < 0.05$).

**Data Availability Statement:** All relevant data are within the manuscript and its Supporting Information files.

**Funding:** The author(s) received no specific funding for this work.

**Competing interests:** The authors have declared that no competing interests exist.

## Significance

In conclusion, chronic codeine use resulted in testicular degeneration and testosterone suppression, which is attributable to TNF-α/nitric oxide-/oxidative stress-mediated caspase-dependent apoptotic testicular cell death and loss of testicular function.

## Introduction

Maintenance of fertility is of great concern in the male population, particularly with the rise in the cases of infertility due to male factor. Among the risk factors of male infertility are drug abuse and chronic drug use. Drugs have been shown to impair male fertility via hormonal and non-hormonal mechanisms [1]. They have been reported to suppress the hypothalamic-pituitary-testicular axis resulting in reduced biosynthesis of testosterone [2–5], and consequent decline in spermatogenesis. Some cause direct damage to the sperm cells [6, 7] while others block spermatogenesis [8].

Opioids are among the medications most frequently abused worldwide for self-management of pain and recreational purposes [9–11]. Prolong use of prescription-based opioids can lead to misuse, dependence, and mortality due to their rewarding and reinforcing properties [12], which has made the misuse of pharmacological opioids a growing concern. The prevalence of opioid abuse and misuse among adult opioid users in the United States was 12.8% and 12.5% in 2015 and 2016, respectively [13]. In 2016, about 34.3 million people, accounting for 0.7% of the global population, abused opioids [14].

Chronic use of opioids is associated with androgen deficiency [15–18]. Findings of Rubinstein and Carpenter [15] demonstrated that the highest odds of androgen deficiency are associated with opioids that maintain the most stable serum drug concentrations. Since the biosynthesis of testosterone occurs in a pulsatile manner, it is possible that the nadirs in the concentration of the drug that occurs between doses of short-acting opioids permit testosterone production that is sufficient to maintain the constant circulating concentration of the androgen [15]. Contrary to this, long-acting opioids suppress testosterone synthesis completely [15].

Although opioids can cause testicular toxicity, mechanisms associated with opioid-induced testosterone suppression remain unclear. Studies of Azari *et al.* [19] revealed that tramadol destroyed testicular germinal layer and atrophy of seminiferous tubules. Their study also revealed that tramadol impaired spermatogenesis and led to necrosis of spermatocytes with nuclear changes such as karyolysis and karyorrhexis. Findings of Nna *et al.* [20] corroborate this report. They observed reduced leydig cell, testicular germ cell and epididymal spermatozoa density in tramadol-treated rats.

Opioids differ in structure and lipophilicity [12, 15], hence may show variations in their mode of actions. Codeine has been reported to be the most commonly abused opioids globally [21]. Bakare and Isah [22] also reported a similar prevalence among Nigerian abusers. Despite the high prevalence of codeine abuse, there is a dearth of knowledge on the effect of chronic codeine use on testis and testosterone bioavailability. Although, our previous study observed that chronic codeine administration in rabbits led to enhanced sexual motivation and copulatory locomotor activity, as well as reduced copulatory efficiency, fertility index, and serum level of testosterone [23], the mechanisms associated with codeine-induced decline in testosterone concentration is yet to be elucidated. This study sought to demonstrate the possible mechanisms associated with codeine-induced testosterone suppression.

## Methods

### Drugs and chemicals

Assays were done using standard ELISA kits. All other reagents used, unless otherwise indicated, were of analytical grade and obtained from Sigma Chemical Co, USA.

### Experimental animals

Twenty-one male rabbits of comparable weight and age (10±2 weeks, 950±50g) housed in standard well-ventilated cages (maximum of 2 animals/cage), in the Animal Holdings of the Department of Physiology, Ladoke Akintola University of Technology, Ogbomoso, Nigeria. The rabbits were provided with standard chow and water *ad libitum* and subjected to the natural photoperiod of 12 h light/dark cycle. All animals were cared for humanely according to the conditions stated in the Guide for the Care and Use of Laboratory Animals' of the National Academy of Science (NAS), published by the National Institute of Health. The experimental protocol was in accordance with the US NAS guidelines and ethical approval obtained from the Ethics Review Committee, Ministry of Health, Oyo State, Nigeria, with Reference Number AD13/479/1396.

### Experimental procedure

Animals were randomly allocated into three groups of seven rabbits each and administered the following:

Group I: control rats received 1ml of 0.9% normal saline.

Group II: low-dose codeine, received 4mg/kg b.w of codeine.

Group III: high-dose codeine, received 10mg/kg b.w of codeine.

All administrations were done orally using oro-pharyngeal cannula daily for six weeks. The 4mg/kg dose was based on the Human Equivalent Dose, while the 10mg/kg was obtained from the dose-response curves to obtain a submaximal peak dose. This is as reported in our previous study [23].

Twenty four hours after the last treatment, the over-night-fasted rabbits were weighed and sacrificed via intraperitoneal administration of ketamine (40mg/kg) and xylazine (4mg/kg) [24]. Blood samples were collected via cardiac puncture, centrifuged at 3000rpm for 10 minutes, and the serum separated for hormonal assay. Both testes were excised. Surrounding structures and adipose tissues were trimmed off, and the paired testicular weight of each animal was determined using a sensitive electronic scale. The left testes were kept in sample bottles containing phosphate buffer for further biochemical assay [25, 26]. The testes were homogenized in the buffer, and the homogenate was centrifuged at 10,000rpm for 15 minutes at 4˚C to obtain the supernatant for biochemical assay. The harvested right testes were fixed in Bouin's solution for histological processing [27, 28].

### Estimation of testicular enzymes

The testicular activity of alkaline phosphatase (ALP) was determined using a standard kit (Teco Diagnostics, USA). Briefly, testicular tissue was homogenized in phosphate buffer. For each sample, 0.5mL of ALP substrate was dispensed into labeled test tubes and equilibrated to 37˚C for 3 minutes. At timed interval, 50μl of each standard, control, and testicular homogenates were added to appropriate test tube. Deionized water was used as sample for reagent blank. The solution was mixed gently and incubated for 10 minutes at 37˚C. Following this

sequence, 2.5mL of ALP colour developer was added at timed interval and mixed thoroughly. The wavelength of the spectrophotometer was set at 590nm zero with reagent blank (wavelength range: 580-630nm). The absorbance of each sample was read and recorded.

The activity of acid phosphatase (ACP) was determined using a standard kit (Pointe Scientific Inc., USA). Immediately after separation of the supernatant, ACP was stabilized by adding 20μl of Acetate Buffer per 1.0ml of supernatant. The solution was mixed and stored in refrigerator until assay was performed. 1.0 ml of reagent was added to appropriately labeled tube, and then 10 μl of L-Tartrate reagent was added and properly mixed. The spectrophotometer was zeroed with water at 405nm, and cuvette temperature set to 37˚C. 100 μl of each sample was added, mixed and incubated for 5 minutes. After incubation, the absorbance was read and recorded every minute for five minutes to determine ΔA/minute. Values (U/L) were obtained by multiplying ΔA/minute by the factor.

The activity of lactate dehydrogenase (LDH) was determined using a standard kit (Agappe Diagnostics Ltd., India). 1000μL of working reagent and 10μL of testicular homogenate was mixed and incubated at 37˚C for 1 minute. The change in colour absorbance (ΔOD/min) was measured every minute for 3 minutes. The activity of the enzyme was calculated using the formula: LDH-P activity (U/L) = (ΔOD/min) x 16030.

The activity of gamma glutamyl transferase (GGT) was determined using a standard kit (Pointe Scientific Inc., USA). The reagents were prepared according to the instruction manual. 1ml of the reagent was pipetted into appropriately labeled tubes: "control", "sample", and pre-incubated at 37˚C for 5 minutes. The spectrophotometer was zeroed with water at 405nm. 100 μl of the homogenate was added, mixed and returned to a thermo cuvette. After 60 seconds, the absorbance was read and recorded every minute for 2 minutes. The mean absorbance difference per minute (Δ Abs./min.) was determined and multiplied by 1158 to obtain results in U/L.

## Estimation of testicular total protein and glucose

Testicular total protein concentration was measured using the biuret method [29], which depends on the presence of peptide bonds in all proteins. These peptide bonds react with $Cu^{2+}$ ions in alkaline solutions to form a coloured product. The absorbance of which is measured spectrophotometrically at 540nm.

Testicular glucose concentration was determined using standard kit (Agappe Diagnostics Ltd., India). 1000μL of the working reagent and 10μL of the testicular homogenate were mixed and incubated for 10 minutes at 37˚C. The absorbance of standard and sample were read against reagent blank, and the concentration of glucose determined by the formula:

$$\text{Glucose concentration (mg/dl)} = \frac{\text{Absorbance of sample}}{\text{Absorbance of standard}} \text{X 100}$$

## Estimation of serum and testicular levels of reproductive hormones

Serum and testicular testosterone were determined using ELISA kit (Monobind Inc. USA; product number: 4806-300A), serum oestrogen was determined using ELISA kit (Monobind Inc. USA; product number: 4925-300A), serum FSH was assayed using ELISA kit (Monobind Inc. USA; product number: 506-300A), and serum LH was assayed using ELISA kit (Monobind Inc. USA; product number: 625-300A). The hormones were assayed as previously documented [26, 27]. Briefly, for serum testosterone assay, 10μL of standards, control and serum samples were dispensed into their respective wells. 100 μL Testosterone-HRP conjugate was added to each well. Substrate blank was dispensed into well A1. The wells were then covered with foil. Then incubation was carried out for 1 hour at room temperature, the foil was

removed and well contents aspirated. Then 300 μL diluted wash solution was used three times to wash the wells. The soak time between each wash cycle was more than 5 seconds. The remaining fluid was carefully removed by tapping the strips on tissue paper. 100 μL of TMB substrate solution was added into all wells. The wells were incubated for 15 minutes at room temperature in the dark. 100 μL stop solution was dispensed into all wells in the same order and the same rate as for the substrate. The absorbance of the specimen was read at 450nm within 20 minutes after addition of the stop solution.

Testicular testosterone measurement was determined similarly; however, the testes supernatant obtained from testicular homogenate was used instead of the serum.

Similarly, determination of serum oestrogen was by dispensing 25μL of standards, control and serum samples into their respective wells. 50μL of E2 Biotin reagent was added to all wells and incubated for 30 minutes at 37˚C after which 50μL of enzyme conjugate was added to each well. Well A1 was left for substrate blank. The wells were covered with foil. Incubated of the wells was for 90 minutes at room temperature. The foil was removed and well contents aspirated. 300 μL diluted wash solution was used to wash the wells three times. The soak time between each wash cycle was more than 5 seconds. The remaining fluid was carefully removed by tapping the strips on tissue paper. 100μL of TMB substrate was added into all wells. Incubation of the wells was carried out in the dark at 37˚C for 30 minutes. 50 μL of stop solution was then dispensed into all wells in the same order and time interval as the substrate. The absorbance of the specimen was read at 450nm within 20 minutes after the addition of the stop solution.

For FSH assay, 50μL of standards, control and serum were dispensed into their respective wells. 100 μL of the conjugate was added to each well. Substrate blank was dispensed into well A1. The wells were covered with foil. Incubation of the wells was carried out for 1 hour at 37˚C. After 1 hour the foil was removed and well contents aspirated, wells washed three times with 300 μL of diluted wash solution. The soak time between each wash cycle was more than 5 seconds. The remaining fluid was carefully removed by tapping the strips on tissue paper. 100 μL of TMB substrate was added into all wells. The wells were incubated for 15 minutes at room temperature in the dark. 50μL of stop solution was dispensed into all wells in the same order and rate as for the substrate. The absorbance of the specimen was read at 450nm.

For LH assay, 50μL of standards, control and serum samples were dispensed into their respective wells in duplicates. 100 μL conjugate was added to each well. Substrate blank was dispensed into well A1. The wells were left to incubate for 1 hour at room temperature. Contents of wells were aspirated after 1 hour, and the wells were washed three times using 300 μL diluted wash solution. The remaining fluid was carefully removed by tapping the strips on tissue paper. 100μL of TMB substrate solution added into all wells. The wells were incubated for 15 minutes at room temperature in the dark. 100μL of stop solution was dispensed into all wells in the same order and rate as for the substrate. The absorbance of the specimen was done at 450nm.

The concentrations of the hormones were obtained from the absorbance values using standard curves obtained.

## Estimation of testicular oxidative markers, anti-oxidants, 8OHdG and caspase 3

Lipid peroxidation concentration determination is by measuring the thiobarbituric acid reactive substances (TBARS) produced during lipid peroxidation as previously documented [30]. This method is dependent on the reaction between 2- thiobarbituric acid (TBA) and malondialdehyde, an end product of lipid peroxidation. On heating in acidic pH, a pink chromogen

complex ([TBS] 2-malondialdehyde adduct) is formed and measured by its absorbance at 532 nm. Briefly, 200µL of the sample was deproteinised with 500µL of Trichloroacetic acid (TCA) and centrifuged at 3000rpm for 10minutes. 1mL of 0.75% TBA was added to 0.1mL of the supernatant and boiled in a water bath for 20minutes at 100˚C and cooled with ice water. The absorbance of the sample/standard was read at 532nm with a spectrophotometer against the blank. The concentration of TBARS generated was extrapolated from the standard curve.

Hydrogen peroxide ($H_2O_2$) generation was determined by the method described by Nour-oozzadeh *et al.* [31]. The method uses a colour reagent that contains xylenol orange dye in an acidic solution with sorbitol and ferrous ammonium sulphate that reacts to produce a purple colour in proportion to the concentration of $H_2O_2$. The assay mixture was thoroughly vortex-mixed until it foamed. A pale pink colour complex was generated after incubation for 30 minutes at room temperature. The absorbance was read against blank (distilled water) at 560nm wavelength. The concentration of hydrogen peroxide generated was extrapolated from the standard curve.

Myeloperoxidase activity was determined as previously documented by Desser *et al.* [32]. Myeloperoxidase catalyses the oxidation of guaiacol to oxidised guaiacol in the presence of hydrogen peroxide. Guaiacol in its oxidised form has a brown colour which is measured photometrically at 470nm wavelength. The concentration of oxidised guaiacol produced in the reaction is proportional to the colour intensity produced.

Superoxide dismutase (SOD) activity was determined as previously reported [33, 34]. Briefly, 1mL of the sample was diluted in 9mL of distilled water to make a 1 in 10 dilutions. An aliquot of 0.2mL of the diluted sample was added to 2.5mL of 0.05M carbonate buffer (pH 10.2) for spectrophotometer equilibration and the reaction started by the addition of 0.3mL of freshly prepared 0.3mM adrenaline to the mixture which was quickly mixed by inversion. The reference cuvette contained 2.5mL buffer, 0.3mL of the substrate (adrenaline) and 0.2mL of water. The increase in absorbance was monitored every 30 seconds for 150 seconds 480nm.

Testicular catalase activity was determined as previously documented [35] with some modifications. Briefly, 1ml of the supernatant of the testicular homogenate was mixed with 19ml of diluted water to give a 1:29 dilution of the sample. The assayed mixture contained 4ml of $H_2O_2$ solution (800µmoles) and 5ml of phosphate buffer in a 10ml flat bottom flask. 1ml of properly diluted enzyme preparation was mixed with the reaction mixture by a gentle swirling motion at 37˚C room temperature. 1ml of the reaction mixture was withdrawn and blown into 2ml dichromate/acetic acid reagent at 60 seconds intervals. The $H_2O_2$ contents of the withdrawn sample were determined. Catalase levels in the sample are determined by comparing absorbance at 653 nm to that of a certified catalase standard.

The method of Beutler *et al.* [36] was used in estimating the level of reduced glutathione (GSH). An aliquot of the sample was deproteinized by the addition of an equal volume of 4% sulfosalicylic acid, which was centrifuged at 4,000rpm for 5 minutes. After that, 0.5ml of the supernatant was added to 4.5ml of Ellman's reagent. A blank was prepared with 0.5ml of the diluted precipitating agent and 4.5ml of Ellman's reagent. Reduced glutathione, GSH, is equal to the absorbance at 412nm.

Testicular glutathione peroxidase (GPx) activity was determined according to the method of Rotruck *et al.* [37] with some modifications. Briefly, the reaction mixture containing the sample was incubated at 37˚C for 3 minutes after which 0.5mL of 10% trichloroacetic acid (TCA) was added and after that centrifuged at 3000rpm for 5 minutes. To 1mL of each of the supernatants, 2mL of phosphate buffer and 1mL of 5'-5'- dithiobis-(2-dinitrobenzoic acid (DTNB) solution was added, and the absorbance was read at 412nm against a blank. Glutathione peroxidase activity was observed by plotting the standard curve, and then the concentration of the remaining GSH extrapolated from the curve.

Testicular Glutathione-S-transferase activity was determined using the method described by Habig *et al.* [38]. The method is based on the relatively high activity of glutathione-S-transferase with 1-chloro-2,4,-dinitrobenzene as the second substrate. The prepared medium of the assay and the reaction was allowed to run for 60 seconds, then the absorbance was read against the blank at 340nm. The temperature was put at approximately 37˚C.

AGE was determined using a standard ELISA kit (Elabscience Biotechnology Co., Ltd, USA) following the manufacturer's manual. The standard working solution was added to the first two columns: Each concentration of the solution was added in duplicate, to one well each, side by side (50 uL for each well). Samples were added to the other wells (50 uL for each well). Immediately, 50μL of Biotinylated Detection Ab working solution was added to each well. The plate was then covered with the provided sealer in the kit and incubated for 45 minutes at 37˚C. Care was made to ensure that the solutions were added to the bottom of the micro ELISA plate well. Also, avoidance of touching the inside wall and causing foaming as much as possible was ensured. The solution aspirated from each well, and 350 uL of wash buffer added to each well. These were soaked for 1–2 minutes, and the solution was aspirated from each well, then patted dry against clean absorbent paper. The wash step was repeated three times. 100 μL of HRP Conjugate working solution was added to each well and covered with the plate sealer then incubated for 30 minutes at 37˚C. The solution from each well was aspirated, and the wash process was repeated five times. Then 90 μL of Substrate Reagent was added to each well and covered with a new plate sealer then incubated for another 15 minutes at 37˚C. The plate was protected from light. After this, 50 μL of stop solution was added to each well. The value of the light density of each well was determined with a microplate reader set to 450 nm.

8OHdG level and cysteine-aspartic proteases 3 (caspase 3) activities were determined as described for AGE using ELISA kits (Elabscience Biotechnology Co., Ltd, USA. Procedures were as stated in the manufacturer's manual.

## Assessment of testicular nitric oxide and TNF-α

Testicular nitric oxide (NO) was determined using Griess Reaction as previously documented [39]. 100μL of Griess reagent, 300μL of nitrate-containing testicular homogenate, and 2.6ml of deionized water were mixed in a spectrophotometer cuvette and incubated for 30 minutes at room temperature. The blank was prepared by mixing 100μL of Griess reagent and 2.9ml of deionised water. The absorbance of the nitrate-containing sample was measured at 548nm relative to the reference sample.

Testicular level of TNF-α was determined using a standard ELISA kit (Elabscience Biotechnology Co., Ltd, USA). 100 μL of the standard and sample was added to each well, and incubated for 90 min at 37˚C. The liquid was then removed. 100 μL of Biotinylated Detection Ab. Solution was added and incubated for 1 hour at 37˚C. The solution was aspirated and washed 3 times. 100 μL of HRP Conjugate was then added, and incubated for 30 min at 37˚C. The solution was aspirate from each well and washed 5 times. 90 μL of the substrate reagent was added and incubated for 15 min at 37˚C. 50 μL of stop solution was added. The optical density of each well was read at 450 nm immediately, and values were then calculated.

## Assessment of testicular DNA fragmentation

Testicular DNA fragmentation was determined as previously described [40] with some modifications. 5ml each of testicular homogenate supernatant and pellet was treated with 3ml of freshly prepared diphenylamine (DPA) reagent for colour development. The solution was incubated at a temperature of 37˚C for 16 to 24 hours. The absorbance of light green/yellowish-green supernatant was read spectrophotometrically at 620nm. The percentage of

fragmented DNA was determined as the result of the absorbance of homogenate supernatant divided by the sum of the absorbance of the pellet and absorbance of supernatant.

Microplate reader MR 9608 (AZZOTA corporation, New Jersey, USA) and spectrophotometer S23A (Gulfex Medical and Scientific Limited, United Kingdom) were used for the study.

## Histology

Sections of the fixed tissues were obtained from paraffin blocks and stained with Haematoxylin and Eosin (H&E) [41]. Stained sections of the testis were observed under a digital light microscope at various magnifications and photographs were taken.

## Statistical analysis

Data were expressed as mean ± SD. The data comparisons made use of one-way analysis of variance (ANOVA) followed by Tukey's post hoc test for pairwise comparison. Pearson's bivariate correlation was done to assess the correlation between selected indices of testicular functions and 8-OHdG, nitric oxide and caspase 3. Statistical Package for Social Sciences (SPSS, version 16) was used for the analysis. Statistical significance was set at $p < 0.05$.

## Results

### Effect of chronic codeine use on body weight, testicular weight and relative testicular weight

Table 1 represents the body and testicular weights of the treated and control groups. Oral administration of codeine for six weeks did not statistically affect the body weight gain. However, oral administration of 4mg/kg and 10mg/kg bodyweight of codeine showed 50% and 68% decline in paired testicular weight, respectively. The data on paired testicular weight presented in Table 1 showed a dose-dependent decrease in the paired testicular weight of the treated rabbits. The relative testicular weight determined as the percentage of the ratio between the paired testicular weight and body weight also showed significant differences across the groups. The decline in the relative testicular weight for codeine-treated rabbits at 4mg/kg and 10mg/kg body weight was 47% and 66% respectively below the control rabbits.

### Effect of chronic codeine use on testicular enzymes

Fig 1 shows the activities of testicular enzymes (alkaline phosphatase (ALP), acid phosphatase (ACP), lactate dehydrogenase (LDH) and γ-glutamyl-transferase (GGT) in control and experimental rabbits. The activities of these enzymes were observed to be significantly raised in the treated rabbits when compared with the control rabbits ($p < 0.05$). Oral administration of codeine for six weeks at 4mg/kg body weight led to a 29% rise in ALP, while at 10mg/kg body

**Table 1. Effect of chronic codeine use on body weight, testicular weight and relative testicular weight.**

| Group | Initial BW (g) | Final BW (g) | Difference (g) | Difference (%) | PTW (g) | PTW/BWX 100 |
|---|---|---|---|---|---|---|
| Control | 872.00±69.63 | 1379.1±54.51 | 507.05±32.93 | 58.6257±7.57 | 2.84±0.19 | 0.21±0.01 |
| Low-dose codeine | 866.62±60.21 | 1353.8±71.95 | 487.18±26.15 | 56.3971±4.24 | 1.40±0.55* | 0.11±0.04* |
| High-dose codeine | 818.91±46.69 | 1305.2±64.86 | 486.32±26.69 | 59.4614±3.22 | 0.91±0.38* | 0.07±0.03* |

Values represent mean±SD for 7 replicates in each group. BW, body weight of rats; PTW, paired testicular weight.

*p < 0.05 vs control

#p < 0.05 vs low-dose codeine

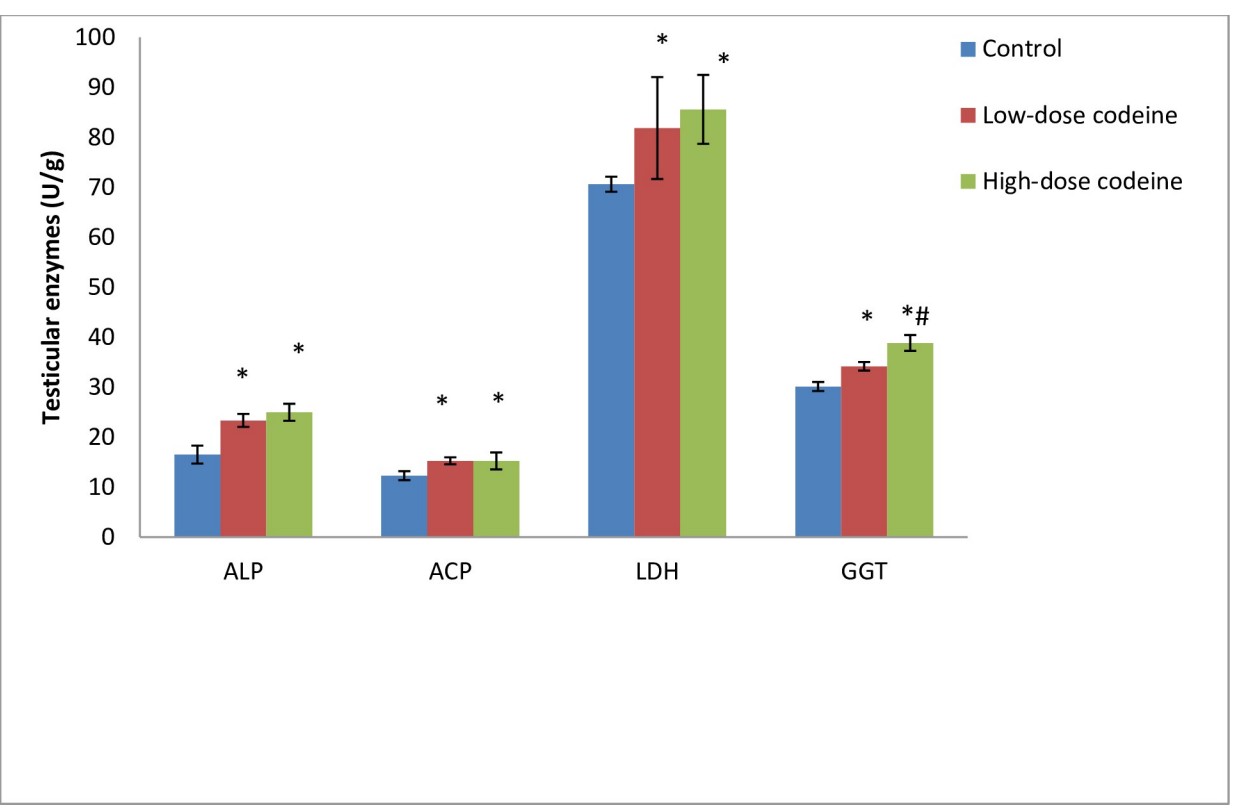

**Fig 1. Effect of chronic codeine use on testicular enzymes activities.** *p < 0.05 vs control, #p < 0.05 vs low-dose codeine.

weight caused a further rise by 6.5%. Similarly, oral codeine administration at 4mg/kg body weight led to a rise in ACP, LDH and GGT by 19%, 13.7%, and 11.8% respectively. At 10mg/kg body weight, codeine administration further increased the activities of LDH, and GGT by 4.4% and 12% respectively, and decreased ACP activity by 0.2% when compared to rabbits who had 4mg/kg body weight of codeine. Although the difference observed in the treated groups were statistically similar, there was a significant difference in GGT activities of both treated groups.

## Effect of chronic codeine use on testicular total protein and glucose

Fig 2 shows testicular levels of total protein and glucose. Although oral administration of codeine at 4 and 10mg/kg body weight led to a rise in total testicular protein by 3.8% and 7.5% respectively, and a similar increase in testicular glucose by 12.9% and 18.6% respectively when compared with the control; these changes were not statistically significant across all groups ($p = 0.153$ and $p = 0.06$ for protein and glucose respectively).

## Effect of chronic codeine use on serum and testicular male reproductive hormones

Fig 3 shows the influence of chronic oral administration of codeine on serum concentrations of testosterone, LH, and FSH, and testicular level of testosterone in rabbits. Circulatory concentrations of LH and FSH in codeine-treated animals significantly increased when compared with those of the control animals ($p = 0.003$ for LH and $p = 0.000$ for FSH). However, oral administration of codeine at 4 and 10mg/kg body weight significantly suppressed circulatory

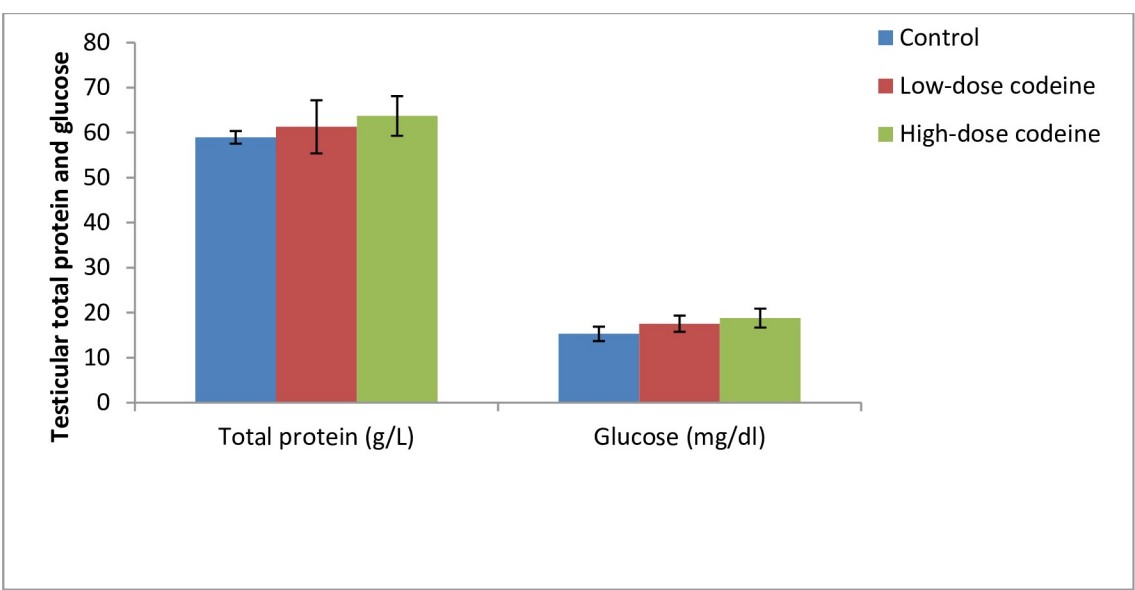

**Fig 2. Effect of chronic codeine use on testicular total protein and glucose.** $^*p < 0.05$ vs control, #p $< 0.05$ vs low-dose codeine.

testosterone levels by 24% and 34% respectively when compared with the control ($p = 0.000$). Similarly, codeine treatment at 4 and 10mg/kg body weight led to a significant dose-dependent decline in the testicular concentration of testosterone by 24.5% and 34.7% when compared with the control ($p = 0.000$).

Fig 4 shows the values of hormonal fertility indices. The results showed that the values of serum testosterone/luteinizing hormone (T/LH) and testosterone/oestrogen (T/E$_2$) were

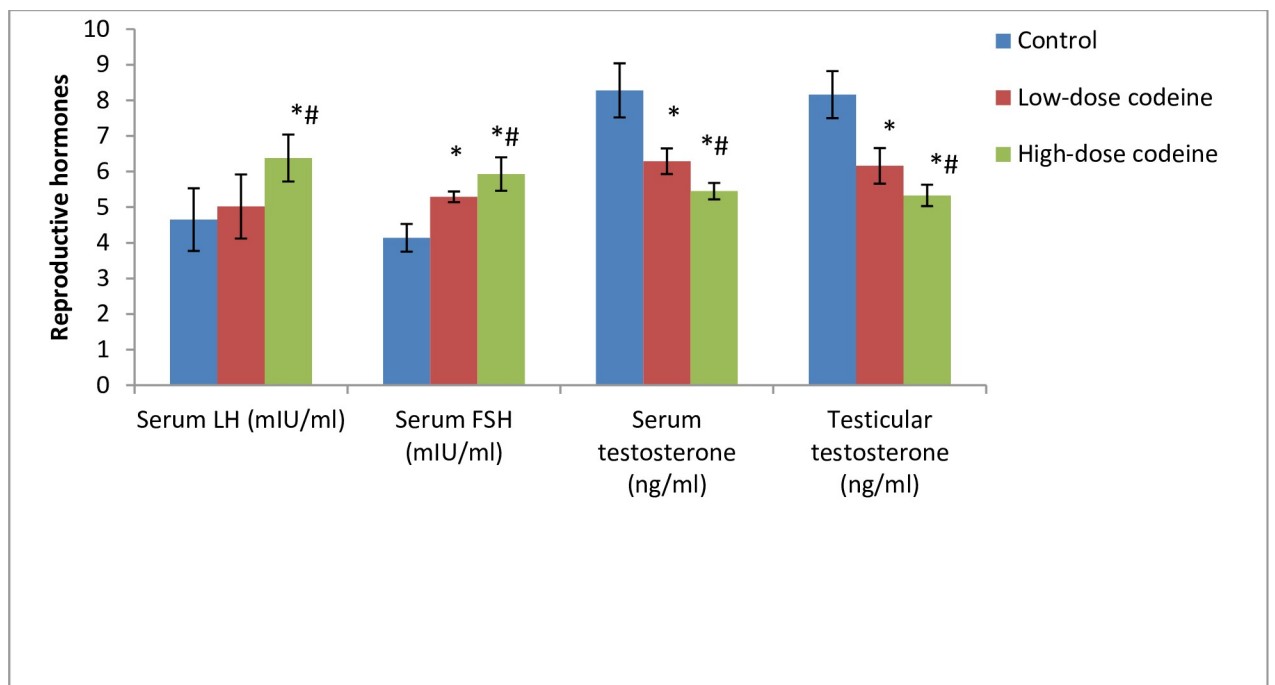

**Fig 3. Effect of chronic codeine use on male reproductive hormones.** $^*p < 0.05$ vs control, #p $< 0.05$ vs low-dose codeine.

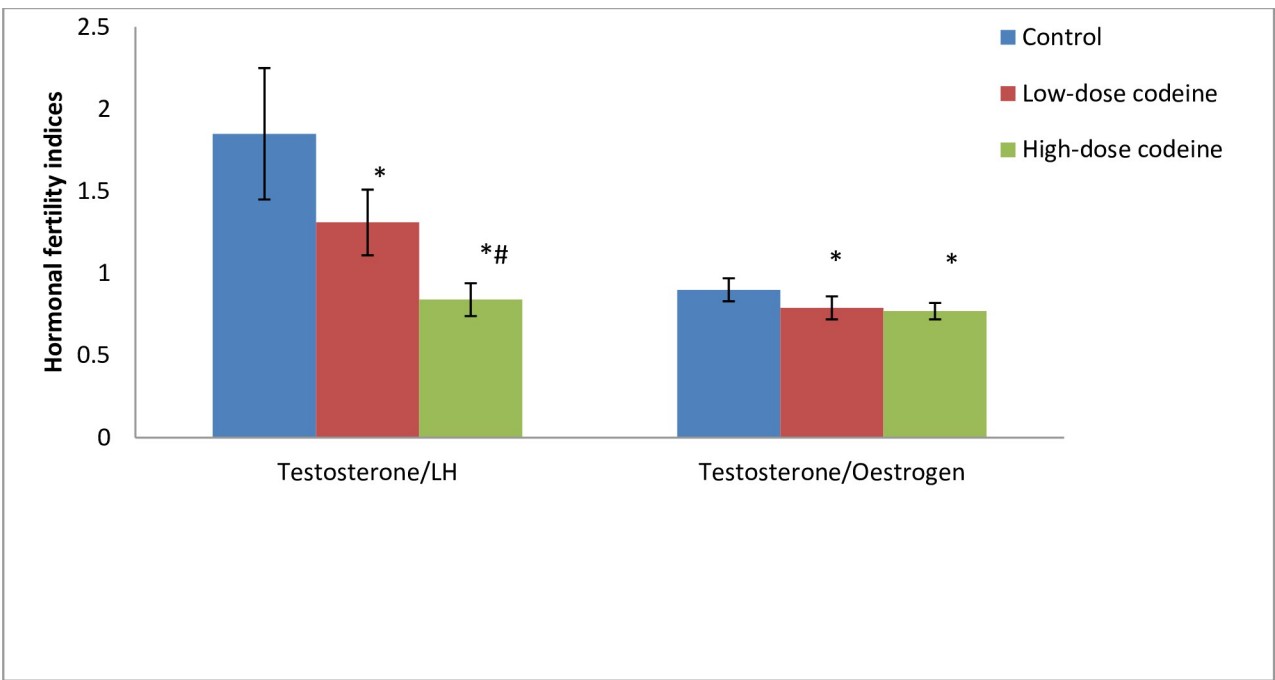

**Fig 4. Effect of chronic codeine use on hormonal fertility indices.** $^*$p $<$ 0.05 vs control, #p $<$ 0.05 vs low-dose codeine.

significantly reduced by codeine exposure. Low dose codeine at 4mg/kg body weight caused 29% and 12% decline in T/LH and T/E$_2$, respectively when compared with the control. Administration of high dose codeine at 10mg/kg body weight caused further decline in these indices by 35% and 2.5% respectively.

## Effect of chronic codeine use on testicular oxidative markers and enzymatic anti-oxidants

Oxidative damage induced by oral administration of codeine in testicular tissue resulted in significant dose-dependent rise in the levels of lipid peroxidation index (malondialdehyde, MDA), oxidative protein denaturation index (AGE), hydrogen peroxide and myeloperoxidase (Fig 5). On the other hand, codeine treatment significantly lowered the activities of SOD, catalase, GSH, GPx and GST. These reductions were also dose-dependent (Fig 6).

## Effect of chronic codeine use on testicular DNA damage, nitric oxide and apoptosis

Compared with the control group, testicular oxidative DNA damage significantly increased in the codeine-treated groups (Fig 7). Low-dose codeine at 4mg/kg body weight led to 27% increase in testicular oxidative DNA damage, while high-dose codeine at 10mg/kg body weight led to 41% increase in testicular oxidative DNA damage when compared with the control. Also, a significant increase in the testicular levels of TNF-α and nitric oxide was observed in codeine-treated rabbits (Fig 8). The rise in TNF-α and nitric oxide seen was dose-dependent. Oral administration of codeine for six weeks at 4 and 10mg/kg body weight led to 48.9% and 53.3% rise in testicular concentration of TNF-α respectively, and 35.6% and 55.7% increase in testicular nitric oxide level respectively.

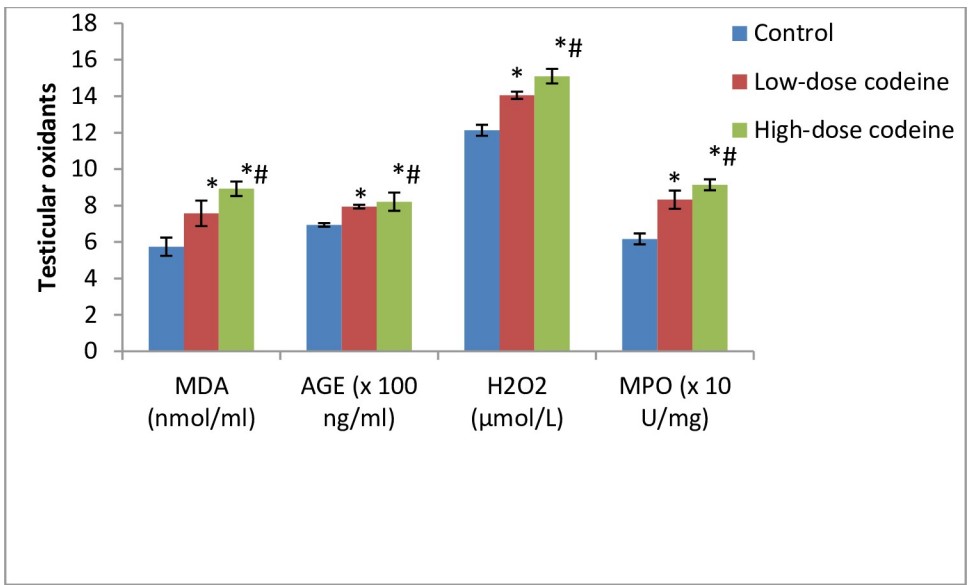

**Fig 5. Effect of chronic codeine use on testicular oxidative stress markers.** $^{*}$p $< 0.05$ vs control, #p $< 0.05$ vs low-dose codeine.

Testicular DNA fragmentation index increased with codeine use in a dose-dependent manner (Fig 9). The assessment of the role of apoptosis was by measuring the activity of caspase 3. Codeine treatment was observed to cause caspase-mediated apoptosis in a dose-dependent manner (Fig 10). At 4 and 10mg/kg body weight, oral administration of codeine significantly increased the testicular caspase 3 activity by 66.8% and 81% respectively when compared with the control.

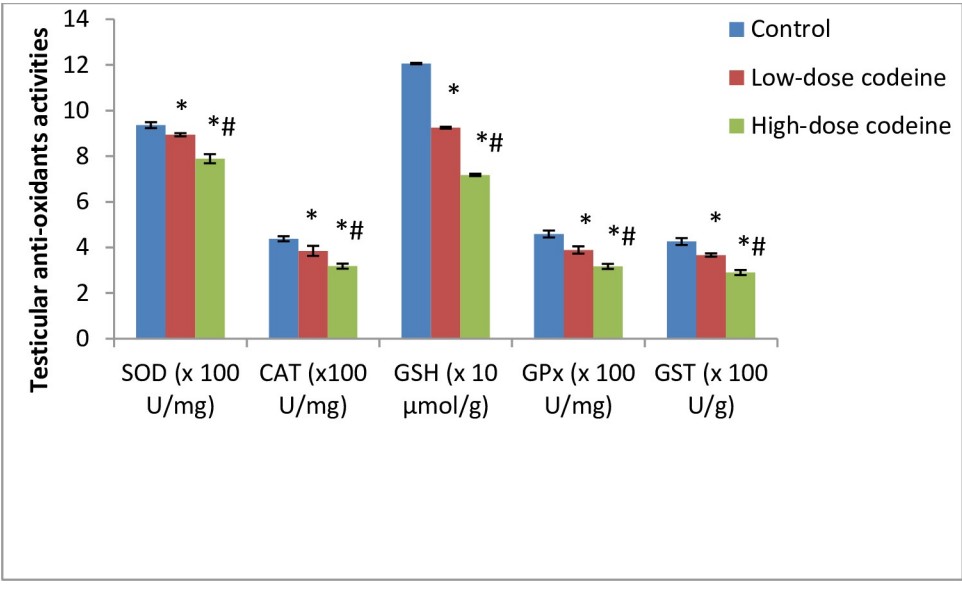

**Fig 6. Effect of chronic codeine use on the activities of testicular enzymatic anti-oxidants.** $^{*}$p $< 0.05$ vs control, #p $< 0.05$ vs low-dose codeine.

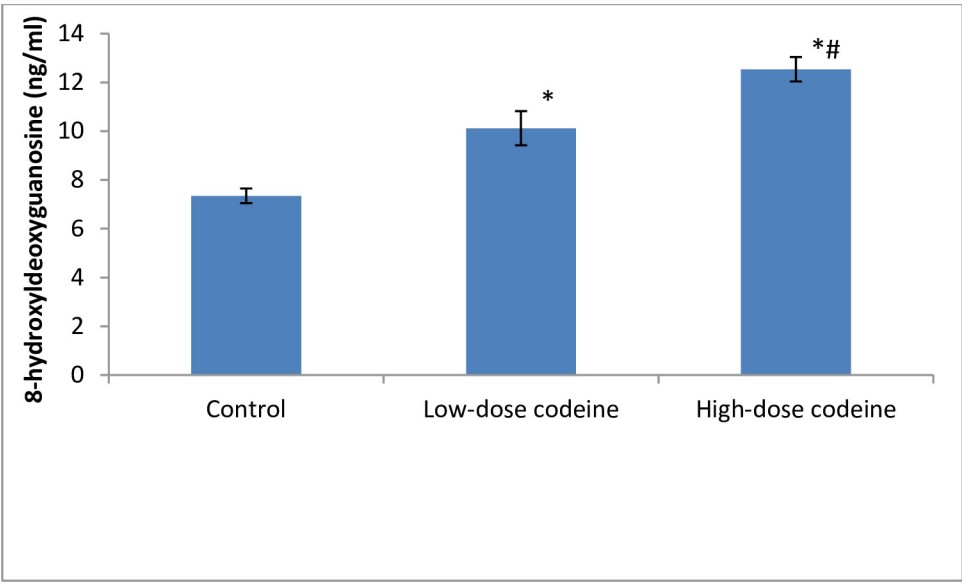

**Fig 7. Effect of chronic codeine use on testicular oxidative DNA damage.** $^*$p $< 0.05$ vs control, #p $< 0.05$ vs low-dose codeine.

## Correlation study

A negative correlation was observed between testicular weight, serum and intra-testicular testosterone, and hormonal fertility indices, and 8-OHdH, nitric oxide and caspase 3 activity (Table 2).

## Histological findings

Findings of histopathological examination supported the present biochemical findings. Photomicrographs (Fig 11) show control rabbits with normal seminiferous tubules containing

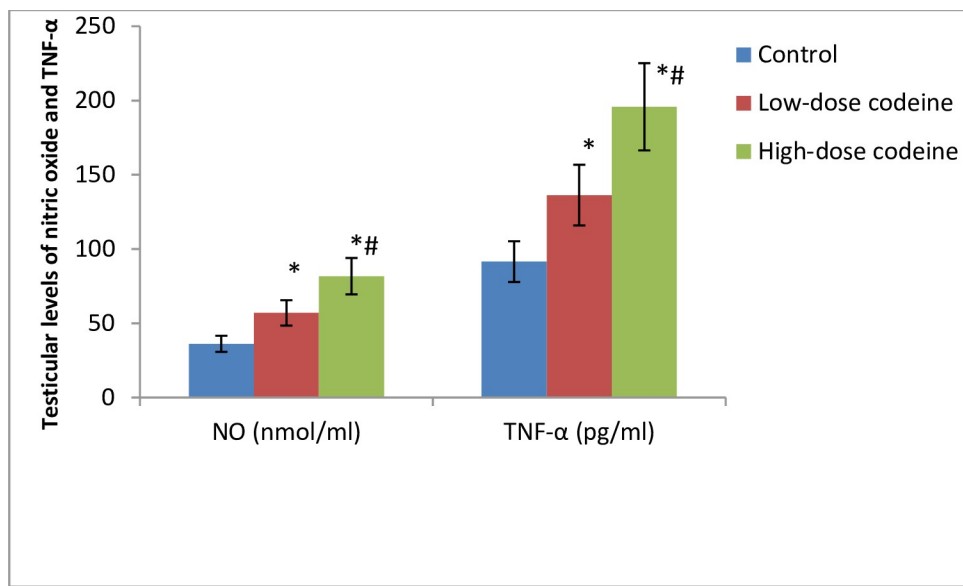

**Fig 8. Effect of chronic codeine use on some inflammatory markers.** $^*$p $< 0.05$ vs control, #p $< 0.05$ vs low-dose codeine.

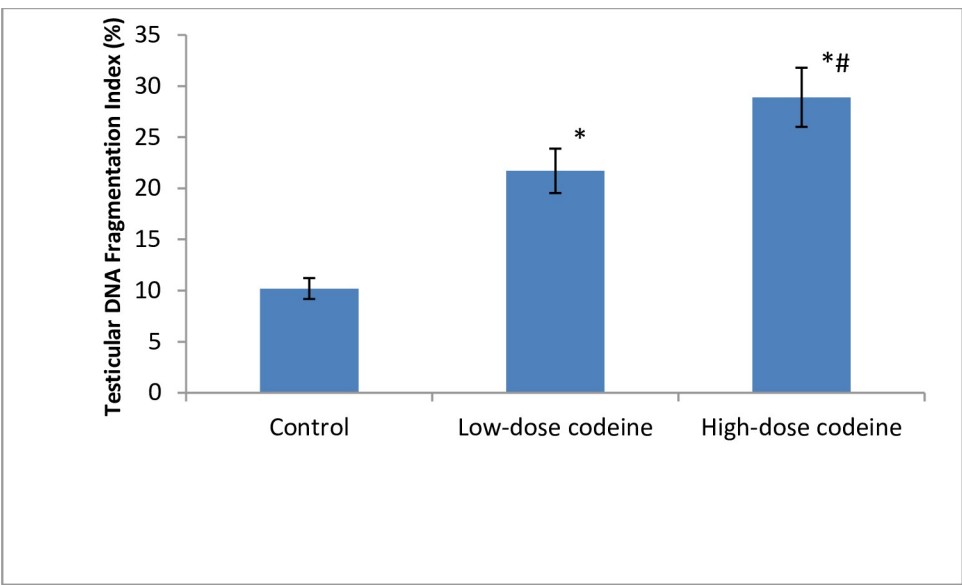

**Fig 9. Effect of chronic codeine use on testicular DNA fragmentation.** $^*p < 0.05$ vs control, #p $<$ 0.05 vs low-dose codeine.

healthy germ cells, including spermatogonia cells and Sertoli cells, and normal maturation stages with the presence of spermatozoa within their lumen. The interstitial spaces show healthy Leydig cells. Oral administration of codeine at 4 and 10mg/kg body weight caused alteration of testicular histoarchitecture. Codeine administration resulted in thickened propria indicative of cessation of spermatogenesis and vascular congestion and associated vacuolation, sloughed germ cells and maturation arrest.

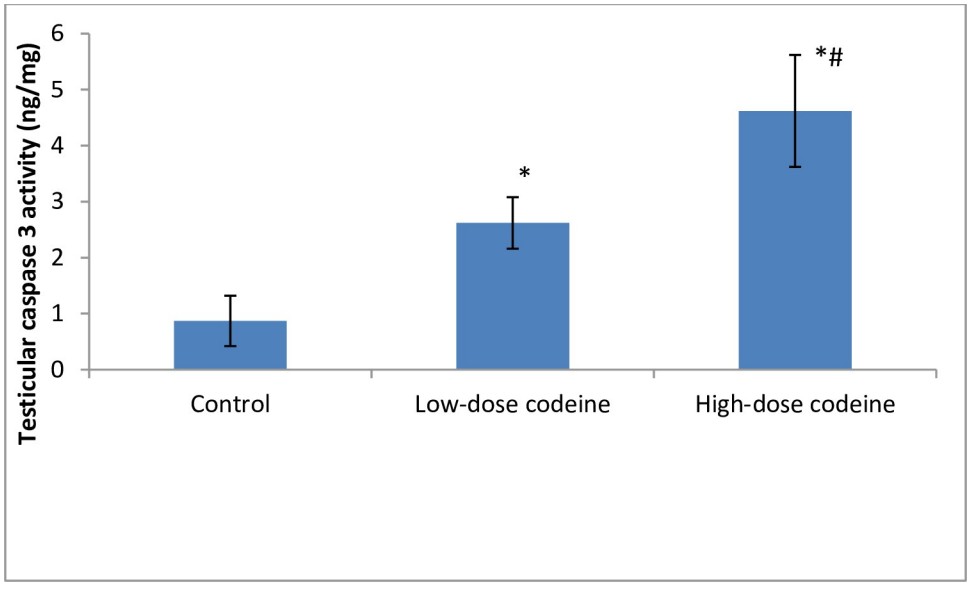

**Fig 10. Effect of chronic codeine use on testicular caspase 3 activity.** $^*p < 0.05$ vs control, #p $<$ 0.05 vs low-dose codeine.

**Table 2. Comparison of some fertility indices with markers of oxidative DNA damage, inflammation, and apoptosis.**

|  | 8-OHdG | Nitric oxide | Caspase 3 activity |
|---|---|---|---|
| **Paired testicular weight** | -0.836 | -0.840 | -0.749 |
| **Relative testicular weight** | -0.823 | -0.824 | -0.713 |
| **Testicular testosterone** | -0.864 | -0.825 | -0.832 |
| **Serum testosterone** | -0.894 | -0.828 | -0.824 |
| **Testosterone/LH** | -0.803 | -0.781 | -0.731 |
| **Testosterone/Oestrogen** | -0.624 | -0.564 | -0.498 |

Values represent mean±SD for 7 replicates in each group. BW, body weight of rats; PTW, paired testicular weight.

$^*$p < 0.05 vs control, #p < 0.05 vs low-dose codeine

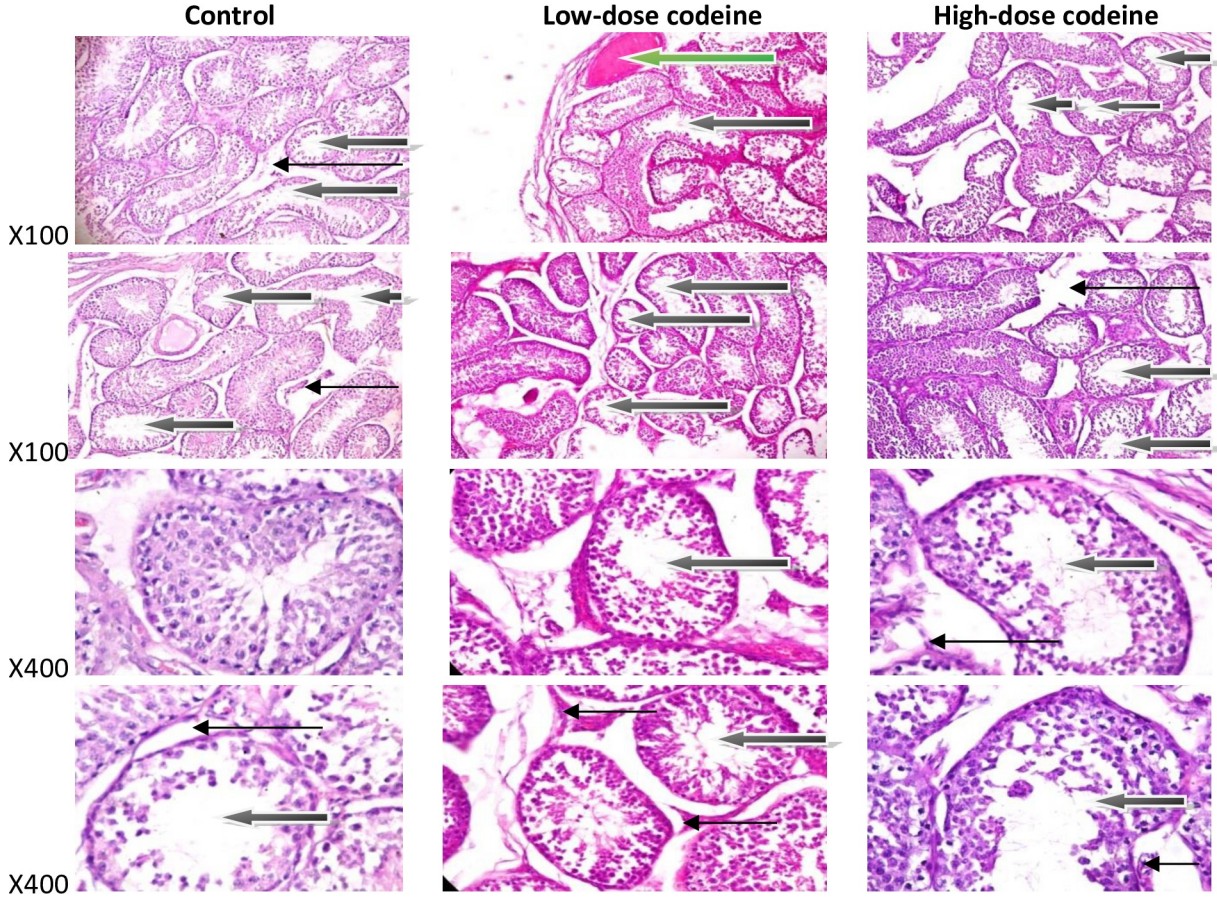

**Fig 11. Photomicrographs of testicular sections stained by Haematoxylin and Eosin.** The control shows some normal seminiferous tubules containing normal germ cells including spermatogonia cells and sertoli cells, and normal maturation stages with presence of spermatozoa within their lumen(white arrow). There very few tubules show maturation arrest at secondary stage of maturation (black arrow). The interstitial spaces show normal leydig cells (slender arrow). The rabbits on low-dose codeine show very poor architecture. The seminiferous tubules show double cell layer or thickened propria indicative of cessation of spermatogenesis. They exhibit vacuolation, sloughed germ cells and maturation arrest and show giant cells (black arrow). There is evidence of vascular congestion (green arrow). The interstitial spaces appear normal (slender arrow). The animals on high-dose codeine show very poor architecture. The seminiferous tubules show double cell layer or thickened propria indicative of cessation of spermatogenesis. They exhibit vacuolation, sloughed germ cells and maturation arrest and show giant cells (black arrow). The interstitial spaces appear normal (slender arrow).

## Discussion

The abuse of codeine, a 3-methylmorphine, remains a global health threat. This drug molecule has been reported to suppress testosterone production [23]. However, this study is the first to demonstrate the toxic effect of codeine on the testis and testicular function via oxidative- and inflammation-mediated caspase-dependent apoptosis. Ranging from a reduction in testicular weight, impaired testicular functions, spermatogenesis cessation, reduced hormonal fertility indices, impaired steroidogenesis, as well as induction of testicular oxidative damage, inflammation, and caspase-mediated apoptosis observed in our study, the need to address the menace of opioid abuse would go a long way in tackling the challenges posed by drug-induced infertility.

The reduction in testicular weight observed in this study is a pointer to codeine-induced alteration in metabolic process that could be associated with excessive tissue break down [42, 43]. Studies have reported the possibility of assessing organ toxicity by the organ weight [44, 45]. Though there was a similar body weight change in the treated and control animals, the decrease in testicular weight, a valuable index of reproductive toxicity, seen in the codeine-treated rabbits indicates that the drug molecule has a toxic effect on the testis and consequent depression of testicular metabolism and function. The reduction in testicular weight following codeine use explains the associated reduction of testosterone bioavailability and disrupted spermatogenesis observed in this study. The decline in the circulatory concentrations of testosterone seen in codeine treatment aligns with previous reports [15–18]. The decline in serum and testicular levels of androgen and cessation of spermatogenesis observed in codeine-treated animals could be partly due to codeine-induced loss of testicular tissues, and impairment of testicular metabolism and function.

Codeine administration induced oxidative stress in the rabbit testis, evident by enhanced lipid peroxidation, oxidative protein denaturation and compromised enzymatic anti-oxidant defence system revealed by decreased glutathione systems with a concomitant reduction in superoxide dismutase and catalase. These findings agree with the report of Nna and Osim [46] that demonstrated the oxidative effects of tramadol on rat testis. Oxidative stress is as a result of an imbalance between reactive oxygen species (ROS) generation and intracellular ROS scavenging capacity. Although the increase in testicular oxidative stress has been reported following use of opioid-like tramadol [46], no study has reported whether or not codeine shares the same activity. The testes are quite prone to ROS-induced injury and peroxidative damage consequent to their high concentrations of polyunsaturated fatty acids and low antioxidant capacity [47].

GSH scavenges ROS by forming oxidized glutathione (GSSG) and other disulfides. Increased oxidative stress promotes the formation and efflux of GSSG, and glutathione reductase (GR) mediates the reduction of GSSG to GSH [48]. This reaction requires the reduced form of nicotinamide adenine dinucleotide phosphate (NADPH), which is provided by glucose-6-phosphate dehydrogenase (G6PD) of the pentose phosphate pathway [48]. Codeine-induced decline in the activity of GSH is indisputably likely consequent to reduced G6PD activity and the resultant decrease in the availability of NADPH. GPx is another constituent of the GSH redox cycle. It is the primary enzyme responsible for hydrogen peroxide ($H_2O_2$) elimination in the testis [49]. GST is well known to catalyse the conjugation of GSH to a range of substrates resulting in detoxification. GPx, together with GST, is vital in $H_2O_2$ metabolism. Thus, the decrease in the activity of GPx and GST, and a rise in the level of $H_2O_2$ observed in the present study in codeine-treated rabbits show that codeine led to peroxide-induced testicular damage. SOD is well known to catalyse the rapid dismutation of superoxide radicals to $H_2O_2$ and dioxygen while catalase converts the formed $H_2O_2$ into water and molecular oxygen.

The significant reduction in the activities of testicular SOD and catalase observed in codeine-treated rabbits in the present study inarguably contributes to the excess $H_2O_2$ seen. Therefore, the overall decline in the glutathione system, SOD and catalase activities triggers robust oxidative stress.

There is growing evidence that oxidative damage permanently occurs to lipids of cellular membranes, proteins, and DNA. In nuclear and mitochondrial DNA, 8-OHdG is a predominant form of free radical-induced oxidative lesions and is a biomarker for oxidative DNA damage [50]. The rise in the level of this biomarker in codeine-treated animals reveals that codeine use promotes testicular oxidative DNA damage.

This study revealed that administration of codeine does not just trigger oxidative damage; it also up-regulates testicular TNF-α, nitric oxide NO, and caspase 3 activity. The overwhelming oxidant stress observed following codeine treatment ultimately excites an inflammatory reaction evident by a rise in testicular TNF-α and NO. It may be secondary to stimulation of NF-kB and enhancement of inducible nitric oxide synthase (iNOS) [51, 52], essential factors in NO production. NO couples with $O_2^-$ to form peroxynitrite ($ONOO_2^-$) which is hugely more reactive and harmful as compared to NO or $O_2^-$ alone. Nitric oxide has also been documented to trigger lipid peroxidation, protein degradation, and DNA damage [52–54]. Hence, the enhancement of testicular NO seen in codeine treatment also accounts for testicular damage. In addition, TNF-α is a key mediator of inflammatory reactions with a broad biological action which include stimulation of NO production [55]. The rise in NO production observed following chronic codeine use is at least in part attributable to increased TNF-α release.

Caspases are significant mediators of apoptosis. They are synthesized as pro-enzymes and activated by internal and/or external stimuli [56]. Caspase 3 is the most critical effector caspase, and the activation of its pathway is a hallmark of apoptosis which can be used as a marker of death cascade [57, 58]. Findings from this study revealed that codeine up-regulated testicular caspase 3 activity. This protease possibly cleaved inhibitor of caspase-activated DNAse I (ICAD) which after then stimulate caspase-activated DNAse I (CAD) [59, 60]. CAD is known to degrade chromosomal DNA within the nucleus and cause chromatin condensation, resulting in DNA fragmentation and cell apoptosis. The rise in testicular caspase 3 activity accounts for the observed testicular DNA fragmentation seen in this study.

Spermatozoa are highly specialized cells that utilize energy for differentiation, development, motility, capacitation, hyper-activation, and acrosome reaction. The energy required for these activities is derived from adenosine triphosphate (ATP) which is primarily produced in the sperm cell via glycolysis and oxidative phosphorylation [61]. It has been established that spermatozoa can survive solely on glycolytic energy [62, 63]. Testicular sertoli cells produce lactate, the preferential energy substrate of germ cells, and pyruvate via the glycolytic pathway using glucose as the major substrate [64], and fructose. LDH catalyzes the inter-conversion of lactate and pyruvate with attendant oxidation/ reduction of NADH to NAD+. This study observed that codeine did not alter testicular concentration of glucose. This infers that the energy substrate for sperm cells remain intact in codeine use.

Activities of testicular enzymes ALP, ACP, LDH and γ-GT are markers of spermatogenesis [65]. They are indices of Sertoli function. The increased activities of ALP and ACP in codeine-administered animals may reflect the release of these non-specific phosphatases from the lysosomes of the degenerating cells and rapid catabolism of the testicular cells [66]. Codeine-induced increased activity of LDH observed in this study reflect testicular degeneration. It is likely due to a compensatory adaptation in an attempt to improve spermatogenesis and testicular regeneration from oxidative damage. γ-GT is a primary index of Sertoli function, and its activity is parallel to Sertoli cell maturation and replication [67]. The marked rise in γ-GT activity in codeine-treated animals in our present study suggests impaired Sertoli cell function

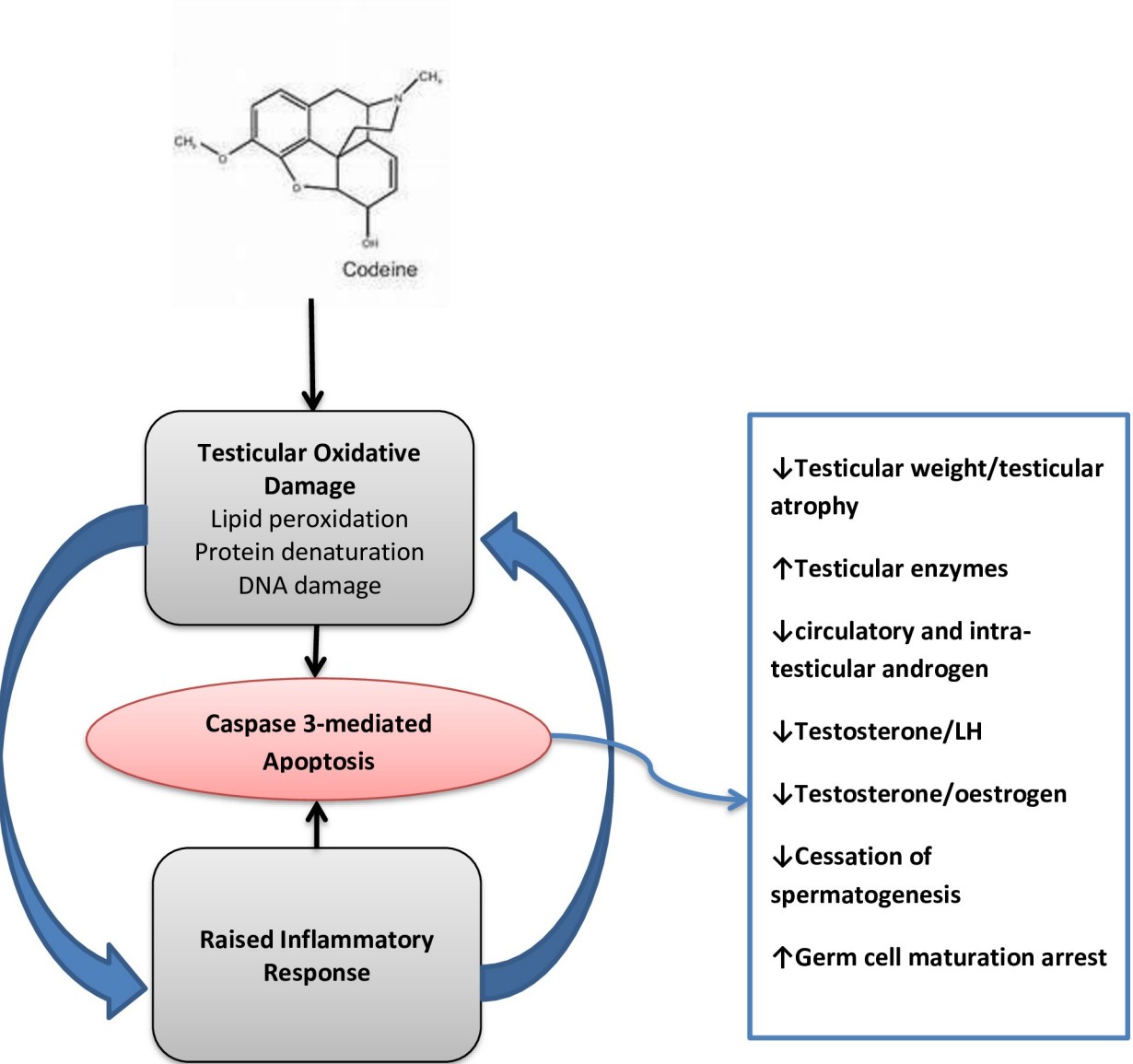

**Fig 12. Graphical abstract showing the proposed mechanism of codeine-induced testicular toxicity.**

and spermatogenesis, manifested by gross testicular atrophy, poor histoarchitecture of the testis, cessation of spermatogenesis, and germ cell maturation arrest.

The dose-dependent diminution in circulatory and intra-testicular concentrations of testosterone in codeine-treated rabbits is obviously due to oxidative damage and caspase-dependent apoptosis of the testes with associated testicular atrophy. Also, suppression of the levels of this androgen explains the rise in LH and FSH observed in codeine-treated animals, possibly via negative pituitary feedback. Spermatogenesis requires a high level of intra-testicular testosterone [68, 69]. In the present study, the cessation of spermatogenesis and the arrest of germ cell maturation seen in codeine-treated animals is a reflection of the reduced testicular level of testosterone. FSH has been documented to stimulate pyruvate and lactate production in Sertoli cells [70], while LDH catalyzes the interconversion of pyruvate and lactate together with the

concomitant inter-conversion of NADH and NAD$^+$. Thus, the raised FSH level in the current study in codeine-treated animals may account for the increased LDH activity seen.

T/LH and T/E$_2$ are hormonal fertility indices. T/LH is a known index of Leydig cell function [71], while high T/E$_2$ is a pointer to fertility [68]. Assessment of T/LH provides more meaningful information than comparisons of testosterone and LH levels separately [71]. In this study, we found that these hormonal fertility indices decreased in codeine-treated rabbits, indicating impaired Leydig cell function, which was manifested by suppression of both circulatory and intra-testicular levels of testosterone.

## Conclusion

Our current study reveals a novel mechanistic pathway for codeine-induced androgen suppression (Fig 12). Codeine depletes testicular glutathione, SOD, and catalase resulting in increased ROS production and inflammatory response which in turn leads to testicular lipid peroxidation, protein denaturation/inactivation, and testicular DNA damage, with consequent caspase-mediated testicular apoptosis and loss of testicular function.

## Supporting information

**S1 Data.**
(XLSX)

## Author Contributions

**Conceptualization:** Roland Akhigbe, Ayodeji Ajayi.

**Data curation:** Roland Akhigbe, Ayodeji Ajayi.

**Formal analysis:** Roland Akhigbe.

**Investigation:** Roland Akhigbe.

**Methodology:** Roland Akhigbe, Ayodeji Ajayi.

**Project administration:** Roland Akhigbe.

**Resources:** Roland Akhigbe.

**Software:** Roland Akhigbe.

**Supervision:** Ayodeji Ajayi.

**Validation:** Ayodeji Ajayi.

**Writing – original draft:** Roland Akhigbe.

**Writing – review & editing:** Ayodeji Ajayi.

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
