## [Decision Letter · Decision Letter 0]

2 Dec 2019

PONE-D-19-27553

Testicular toxicity following chronic codeine administration is via oxidative DNA damage and up-regulation of NO and caspase 3

PLOS ONE

Dear Dr Ajayi,

Thank you for submitting your manuscript to PLOS ONE. After careful consideration, we feel that it has merit but does not fully meet PLOS ONE’s publication criteria as it currently stands. Therefore, we invite you to submit a revised version of the manuscript that addresses the points raised during the review process.

Careful statistical analyses to be performed. Representing the analyzed with relevant units of expression. Describe the methods in detail. Conduct additional experiments to strenthen the molecular basis of toxicity.

We would appreciate receiving your revised manuscript by Jan 16 2020 11:59PM. To enhance the reproducibility of your results, we recommend that if applicable you deposit your laboratory protocols in protocols.io, where a protocol can be assigned its own identifier (DOI) such that it can be cited independently in the future. For instructions see: http://journals.plos.org/plosone/s/submission-guidelines#loc-laboratory-protocols

We look forward to receiving your revised manuscript.

Kind regards,

Suresh Yenugu

Academic Editor

PLOS ONE

Journal Requirements:

1. PLOS requires an ORCID iD for the corresponding author in Editorial Manager on papers submitted after December 6th, 2016. Please ensure that you have an ORCID iD and that it is validated in Editorial Manager. To do this, go to ‘Update my Information’ (in the upper left-hand corner of the main menu), and click on the Fetch/Validate link next to the ORCID field. This will take you to the ORCID site and allow you to create a new iD or authenticate a pre-existing iD in Editorial Manager. Please see the following video for instructions on linking an ORCID iD to your Editorial Manager account: https://www.youtube.com/watch?v=_xcclfuvtxQ

2. Please include your tables as part of your main manuscript and remove the individual files. Please note that supplementary tables (should remain/ be uploaded) as separate "supporting information" files

Additional Editor Comments (if provided):

The testicular toxicity due to drug abuse is well reported in many studies. However, the effect of codeine on testicular toxicity is not well understood and this manuscript provides basic on this aspect.

Additional experiments that provides an in depth analyses on the molecular and cellular mechanisms of codeine toxicity will add more strength to the study.

Statistical analyses are confusing and need to be carefully carried out. In many instances, there appears to be no significance, but it is indicated as significant.

Fructose is the main source of energy for spermatozoa. The rationale in measuring glucose is not provided. How does this fit?

Concrete evidence should be provided as to why testosterone decreases due to codeine treatment? Is it because of the altered levels of FSH / LH or any other cause?

Anti-oxidant enzyme activities should be presented as U/mg protein and not U/ml. Else it will be difficult to normalize the activities / levels between the samples.

Caspase-3 activity should be presented, not the levels (ng/ml).

Reviewers' comments:

Reviewer's Responses to Questions

**Comments to the Author**

1. Is the manuscript technically sound, and do the data support the conclusions?

Reviewer #1: No

2. Has the statistical analysis been performed appropriately and rigorously? 

Reviewer #1: Yes

3. Have the authors made all data underlying the findings in their manuscript fully available?

Reviewer #1: Yes

4. Is the manuscript presented in an intelligible fashion and written in standard English?

Reviewer #1: No

5. Review Comments to the Author

Reviewer #1: The authors investigated in vivo molecular mechanisms of codeine-induced androgen suppression through analyses of hormone, enzyme, oxidative, apoptosis, inflammatory values and histological imaging in rabbits. For the aim, the authors induced three main groups as control, codeine-4 (4 mg/kg) and codeine-10 (10 mg/kg). The authors observed negative effects of codeine treatments on the values through stimulation oxidative stress, inflammation and apoptosis in the testis samples. Topic of the manuscript is interesting for the journal but the manuscript is careless submission. There seem to be areas of considerable scientific confusion in material-method, results and discussion, especially. Before decision on the manuscript, the author should give appropriate answer to major comments at the below.

MAJOR REMARKS

1. There are some syntax and grammatical errors in the manuscript.

2. Abstract is not representing the study. For example, the authors analyzed some antioxidant enzymes in the study, but they were not mentioned in the abstract.

3. The authors expressed that ACP, ALP, LDH and GGT enzymes were statistically further increased in the high dose codeine (10 mg) group as compared to the low dose (4 mg) group. However, they did not increase in the Table 1. In addition, the authors observed testicular increase of total protein and glucose levels by the treatments. However, mean values of the total protein and glucose levels are very similar in the Figure 2 and there are no statistical changes in the Figure 2.

4. In Figures 5 and 6. Oxidation occurs in the protein, lipids and nucleic acids. The authors expressed the testis oxidant and antioxidant values ml instead of mg or g protein. The values should have expressed as mg or g in the manuscript.

5. Some values such as GSH, GPx and MDA are too low according to the reference values. Same problem are present in the remaining values such as glucose and total protein (Please see: Saudi Pharmaceutical Journal Volume 27, Issue 3, March 2019, Pages 326-340; Avicenna J Phytomed. 2019;9(4):374-385).

6. Caspase 3 and 9 are more important enzymes for indicating caspase expression of intrinsic and extrinsic pathways. Therefore, the authors should have also analyzed caspase 9 in the current study at least.

MINOR REMARKS

In abstract

-In line 8. Please use ‘antioxidant’ instead of ‘anti-oxidation’.

- In line 35. Please delete ‘significantly’ words or add p values after the words.

- In line 35 (group were significantly lower than those in LPS group, but still higher than). ‘still higher than’ is not a full sentence. Some words are not present in the sentences.

- In line 37. Please add a space between ‘caspase’ and ‘8’.

- In line 39. Please use ‘methionine selenium can’ instead of ‘selenium can’.

In introduction

- ‘Since the biosynthesis of……..’. The sentence is too long and unclear.

- ‘Contrary to this, long-acting opioids suppress testosterone synthesis more completely’. The sentence is unclear and there are English errors in the sentence.

- Addition of a paragraph on testis, apoptosis and oxidative stress may useful for the section.

In Material and methods

- Most of method sections such as ‘Estimation of testicular enzymes’, ‘Estimation of testicular total protein and glucose’, ‘Estimation of serum and testicular levels of reproductive hormones’ and Estimation of testicular oxidative markers, anti-oxidants, 8OHdG and caspase 3 are too long, although ‘Histology’ section is too short. Please revise the sections.

- Company names and details of the commercial kits were detailed added to the manuscript, but there is no model, company, city and country details of the devices such as ELISA and spectrophotometer in the section. They should be added to the section.

- In ‘8OHdG and caspase three levels were determined as described for AGE using ELISA kits

(Elabscience Biotechnology Co., Ltd, USA) with product numbers E-EL-0028 and E-EL’. The section is too short. Please use ‘caspase 3’ instead of ‘caspase three’ in the section.

In results

- In ‘The decline in the relative testicular weight for codeine-treated rabbits at 4mg/kg and

10mg/kg body weight was 50% and 200% respectively below the control rabbits’. The sentence is not correct and it is conflicting with the results of Table 1. There are no 50% or 200% changes in the table.

- In ‘Effect of chronic codeine use on testicular enzymes’. Please delete one of the double parenthesis in the ‘(GGT))’. SD of LDH in the low codeine group are is too high and I am waiting no difference between control and low codeine group according to the Figure 1 results.

- For Figures 2 and 3 results, please add p values instead of the percentage changes.

-

In discussion

- Most of abbreviations such as reduced glutathione (GSH), nitric oxide (NO) and cysteine-aspartic proteases (caspase) 3. were repeated here, although they we abbreviated in the introduction and material-method sections.

- Discussion section is too short and it lacks focus. The author should concentrate on interpretation of their findings and their relevance to the field of study. The discussion should be used for the interpretation of data and for pointing out the significance of the findings.

In Figures

- In figure 2. Please use antioxidants instead of anti-oxidants.

- In figure 8. There is no unit in the figure.

- In figure 9. Caspase 3 activity should be expressed as mg protein instead of ml.

- Superscripts such as a, b and c are not clear in the figures. For example, In figures 8 and 9, high dose codeine groups should contain b and c superscripts instead of only c.

- Graphical abstract is containing general and brief information. Please extend the graphics. For example, H2O2 is converted to water by GPx and the information should be present in the graphic.

6. PLOS authors have the option to publish the peer review history of their article (what does this mean?). If published, this will include your full peer review and any attached files.

Reviewer #1: Yes: Prof. Dr. Mustafa Nazıroğlu

---

## [Author Response · Author response to Decision Letter 0]

6 Dec 2019

Thanks for the comments. They were found very useful. I have incorporated the corrections per comment. I also attached copies of the ELISA kit manufacturer's manual for caspase and 8OHdG for your perusal. The unit is clearly stated there as ng/ml

Review Comments to the Author

Reviewer #1: The authors investigated in vivo molecular mechanisms of codeine-induced androgen suppression through analyses of hormone, enzyme, oxidative, apoptosis, inflammatory values and histological imaging in rabbits. For the aim, the authors induced three main groups as control, codeine-4 (4 mg/kg) and codeine-10 (10 mg/kg). The authors observed negative effects of codeine treatments on the values through stimulation oxidative stress, inflammation and apoptosis in the testis samples. Topic of the manuscript is interesting for the journal but the manuscript is careless submission. There seem to be areas of considerable scientific confusion in material-method, results and discussion, especially. Before decision on the manuscript, the author should give appropriate answer to major comments at the below.

MAJOR REMARKS

1. There are some syntax and grammatical errors in the manuscript.

Response: Thanks. A thorough grammar check has been done and manuscript has been modified accordingly.

2. Abstract is not representing the study. For example, the authors analyzed some antioxidant enzymes in the study, but they were not mentioned in the abstract.

Response: Thanks. No antioxidant enzyme is listed in the abstract. A total of 9 oxidants and antioxidants were assayed in the present study. Listing them would be cumbersome. Hence, they were referred to as oxidative markers and antioxidants.

3. The authors expressed that ACP, ALP, LDH and GGT enzymes were statistically further increased in the high dose codeine (10 mg) group as compared to the low dose (4 mg) group. However, they did not increase in the Table 1. In addition, the authors observed testicular increase of total protein and glucose levels by the treatments. However, mean values of the total protein and glucose levels are very similar in the Figure 2 and there are no statistical changes in the Figure 2.

Response: Thanks. ACP, ALP, LDH and GGT are presented in Figure 1 and not Table 1. The values of ACP, ALP, and LDH were similar in the codeine 4 and codeine 10 groups but significantly increased when compared to the control. This is evident by the error bars on each. The value of GGT was significantly increased across the groups in a dose-dependent manner.

Testicular protein and glucose were said to be marginal and not statistical different. Hence they are similar. To avoid misinterpretation, the word ‘marginal’ has been replaced with ‘not statistically different ’.

4. In Figures 5 and 6. Oxidation occurs in the protein, lipids and nucleic acids. The authors expressed the testis oxidant and antioxidant values ml instead of mg or g protein. The values should have expressed as mg or g in the manuscript.

Response: Thanks. The unit ‘ng/ml, etc’ is the standard unit and as provided in the manufacturer’s guideline. A copy of the guideline is provided as a supplementary file to the editor.

5. Some values such as GSH, GPx and MDA are too low according to the reference values. Same problem are present in the remaining values such as glucose and total protein (Please see: Saudi Pharmaceutical Journal Volume 27, Issue 3, March 2019, Pages 326-340; Avicenna J Phytomed. 2019;9(4):374-385).

Response: Thanks. The suggested papers have been studied. The values presented here (in our study) are as revealed by standard procedures using standard kits and/or standard laboratory procedures

6. Caspase 3 and 9 are more important enzymes for indicating caspase expression of intrinsic and extrinsic pathways. Therefore, the authors should have also analyzed caspase 9 in the current study at least.

Response: Thanks. Caspase 9 is an initiator and not an effector, but caspase 3 is an effector caspase and the hallmark of apoptosis. It is the primary caspase used as a marker of caspase-mediated apoptosis. Others are just initiators. References are provided in the manuscript. 

MINOR REMARKS

In abstract

-In line 8. Please use ‘antioxidant’ instead of ‘anti-oxidation’.

Response: Thanks. This has been modified.

- In line 35. Please delete ‘significantly’ words or add p values after the words.

Response: Thanks. This has been corrected.

- In line 35 (group were significantly lower than those in LPS group, but still higher than). ‘still higher than’ is not a full sentence. Some words are not present in the sentences.

Response: Thanks. This is not in the manuscript as we do not have an LPS group.

- In line 37. Please add a space between ‘caspase’ and ‘8’.

Response: Thanks. This is not in the manuscript.

- In line 39. Please use ‘methionine selenium can’ instead of ‘selenium can’.

Response: Thanks. This is not in the manuscript.

In introduction

- ‘Since the biosynthesis of……..’. The sentence is too long and unclear.

Response: Thanks. The sentence is brief enough. It means that short acting opioids are characterized by a decline in opioid concentration. This allows sufficient biosynthesis of testosterone (during the fall in opioid concentration). 

This sentence has been modified. 

- ‘Contrary to this, long-acting opioids suppress testosterone synthesis more completely’. The sentence is unclear and there are English errors in the sentence.

Response: Thanks. The sentence has been modified.

- Addition of a paragraph on testis, apoptosis and oxidative stress may useful for the section.

Response: Thanks. Relevant information on codeine and testosterone has been added.

In Material and methods

- Most of method sections such as ‘Estimation of testicular enzymes’, ‘Estimation of testicular total protein and glucose’, ‘Estimation of serum and testicular levels of reproductive hormones’ and Estimation of testicular oxidative markers, anti-oxidants, 8OHdG and caspase 3 are too long, although ‘Histology’ section is too short. Please revise the sections.

Response: Thanks. These sections actually seem quite lengthy because a lot of parameters were assayed. This would be a useful reference for other researchers to aid their methodology.

- Company names and details of the commercial kits were detailed added to the manuscript, but there is no model, company, city and country details of the devices such as ELISA and spectrophotometer in the section. They should be added to the section.

Response: Thanks. These have been included

- In ‘8OHdG and caspase three levels were determined as described for AGE using ELISA kits

(Elabscience Biotechnology Co., Ltd, USA) with product numbers E-EL-0028 and E-EL’. The section is too short. Please use ‘caspase 3’ instead of ‘caspase three’ in the section.

Response: Thanks. ‘Caspase three’ has been replaced with ‘caspase 3’, and details of 8OHdG and caspase 3 assay has been added.

In results

- In ‘The decline in the relative testicular weight for codeine-treated rabbits at 4mg/kg and

10mg/kg body weight was 50% and 200% respectively below the control rabbits’. The sentence is not correct and it is conflicting with the results of Table 1. There are no 50% or 200% changes in the table.

Response: Thanks. This has been re-calculated and corrected

- In ‘Effect of chronic codeine use on testicular enzymes’. Please delete one of the double parenthesis in the ‘(GGT))’. SD of LDH in the low codeine group are is too high and I am waiting no difference between control and low codeine group according to the Figure 1 results.

Response: Thanks. This has been corrected.

There is no difference between the LDH in the low and high codeine groups. However, there is a statistical difference between the control and low codeine group, as well as between the control and high codeine group. This is evident by the error bars.

- For Figures 2 and 3 results, please add p values instead of the percentage changes.

Response: Thanks. The p values have been added

In discussion

- Most of abbreviations such as reduced glutathione (GSH), nitric oxide (NO) and cysteine-aspartic proteases (caspase) 3. were repeated here, although they we abbreviated in the introduction and material-method sections.

Response: Thanks. This has been modified.

- Discussion section is too short and it lacks focus. The author should concentrate on interpretation of their findings and their relevance to the field of study. The discussion should be used for the interpretation of data and for pointing out the significance of the findings.

Response: Thanks. This has been modified.

In Figures

- In figure 2. Please use antioxidants instead of anti-oxidants.

Response: Thanks. This has been corrected 

- In figure 8. There is no unit in the figure.

Response: Thanks. The unit has been added.

- In figure 9. Caspase 3 activity should be expressed as mg protein instead of ml.

Response: Thanks. The unit used is as stated in the manufacturer’s guideline.

- Superscripts such as a, b and c are not clear in the figures. For example, In figures 8 and 9, high dose codeine groups should contain b and c superscripts instead of only c.

Response: Thanks. When using such superscripts, each bar carries JUST ONE superscript, and the interpretation is stated below the figure. ‘Bars of the same parameter carrying different alphabets (a,b,c) are statistically different’ If bar one carries ‘a’, bar 2 carries ‘b’ and bar 3 carries ‘c’, it means that all the group are statistically different from one another. This is as seen in Figures 8 and 9.

If bar one carries ‘a’, bar 2 carries ‘b’ and bar 3 carries ‘b’, it means that groups 2 and 3 are similar but different from group 1. This is as seen in Figure 1. 

If If bar one carries ‘a’, bar 2 carries ‘a’ and bar 3 carries ‘a’, it means that all the groups are similar. This is as seen in Figure 2. 

It is a standard way of showing/representing significance .

- Graphical abstract is containing general and brief information. Please extend the graphics. For example, H2O2 is converted to water by GPx and the information should be present in the graphic.

 Response: Thanks. The graphical abstract summarizes JUST the findings of the present study. Adding details would make it cumbersome and would no longer be a graphical abstract.

---

## [Editor Report · Decision Letter 1]

10 Dec 2019

PONE-D-19-27553R1

Testicular toxicity following chronic codeine administration is via oxidative DNA damage and up-regulation of TNF-α/NO and caspase 3

PLOS ONE

Dear Dr Ajayi,

Thank you for submitting your manuscript to PLOS ONE. After careful consideration, we feel that it has merit but does not fully meet PLOS ONE’s publication criteria as it currently stands. Therefore, we invite you to submit a revised version of the manuscript that addresses the points raised during the review process.

The authors have not provided rebuttal to Academic Editor's comments.

We would appreciate receiving your revised manuscript by Jan 24 2020 11:59PM. To enhance the reproducibility of your results, we recommend that if applicable you deposit your laboratory protocols in protocols.io, where a protocol can be assigned its own identifier (DOI) such that it can be cited independently in the future. For instructions see: http://journals.plos.org/plosone/s/submission-guidelines#loc-laboratory-protocols

We look forward to receiving your revised manuscript.

Kind regards,

Suresh Yenugu

Academic Editor

PLOS ONE

Additional Editor Comments (if provided):

The authors have not provided rebuttal to Academic Editor's comments. Please go through the decision letter sent earlier.

---

## [Author Response · Author response to Decision Letter 1]

10 Dec 2019

Reviewers' comments:

MAJOR REMARKS

1. There are some syntax and grammatical errors in the manuscript.

Response: Thanks. A thorough grammar check has been done and manuscript has been modified accordingly.

2. Abstract is not representing the study. For example, the authors analyzed some antioxidant enzymes in the study, but they were not mentioned in the abstract.

Response: Thanks. No antioxidant enzyme is listed in the abstract. A total of 9 oxidants and antioxidants were assayed in the present study. Listing them would be cumbersome. Hence, they were referred to as oxidative markers and antioxidants.

3. The authors expressed that ACP, ALP, LDH and GGT enzymes were statistically further increased in the high dose codeine (10 mg) group as compared to the low dose (4 mg) group. However, they did not increase in the Table 1. In addition, the authors observed testicular increase of total protein and glucose levels by the treatments. However, mean values of the total protein and glucose levels are very similar in the Figure 2 and there are no statistical changes in the Figure 2.

Response: Thanks. ACP, ALP, LDH and GGT are presented in Figure 1 and not Table 1. The values of ACP, ALP, and LDH were similar in the codeine 4 and codeine 10 groups but significantly increased when compared to the control. This is evident by the error bars on each. The value of GGT was significantly increased across the groups in a dose-dependent manner.

Testicular protein and glucose were said to be marginal and not statistical different. Hence they are similar. To avoid misinterpretation, the word ‘marginal’ has been replaced with ‘not statistically different ’.

4. In Figures 5 and 6. Oxidation occurs in the protein, lipids and nucleic acids. The authors expressed the testis oxidant and antioxidant values ml instead of mg or g protein. The values should have expressed as mg or g in the manuscript.

Response: Thanks. The unit ‘ng/ml, etc’ is the standard unit and as provided in the manufacturer’s guideline. A copy of the guideline is provided as a supplementary file to the editor.

5. Some values such as GSH, GPx and MDA are too low according to the reference values. Same problem are present in the remaining values such as glucose and total protein (Please see: Saudi Pharmaceutical Journal Volume 27, Issue 3, March 2019, Pages 326-340; Avicenna J Phytomed. 2019;9(4):374-385).

Response: Thanks. The suggested papers have been studied. The values presented here (in our study) are as revealed by standard procedures using standard kits and/or standard laboratory procedures

6. Caspase 3 and 9 are more important enzymes for indicating caspase expression of intrinsic and extrinsic pathways. Therefore, the authors should have also analyzed caspase 9 in the current study at least.

Response: Thanks. Caspase 9 is an initiator and not an effector, but caspase 3 is an effector caspase and the hallmark of apoptosis. It is the primary caspase used as a marker of caspase-mediated apoptosis. Others are just initiators. References are provided in the manuscript. 

MINOR REMARKS

In abstract

-In line 8. Please use ‘antioxidant’ instead of ‘anti-oxidation’.

Response: Thanks. This has been modified.

- In line 35. Please delete ‘significantly’ words or add p values after the words.

Response: Thanks. This has been corrected.

- In line 35 (group were significantly lower than those in LPS group, but still higher than). ‘still higher than’ is not a full sentence. Some words are not present in the sentences.

Response: Thanks. This is not in the manuscript as we do not have an LPS group.

- In line 37. Please add a space between ‘caspase’ and ‘8’.

Response: Thanks. This is not in the manuscript.

- In line 39. Please use ‘methionine selenium can’ instead of ‘selenium can’.

Response: Thanks. This is not in the manuscript.

In introduction

- ‘Since the biosynthesis of……..’. The sentence is too long and unclear.

Response: Thanks. The sentence is brief enough. It means that short acting opioids are characterized by a decline in opioid concentration. This allows sufficient biosynthesis of testosterone (during the fall in opioid concentration). 

This sentence has been modified. 

- ‘Contrary to this, long-acting opioids suppress testosterone synthesis more completely’. The sentence is unclear and there are English errors in the sentence.

Response: Thanks. The sentence has been modified.

- Addition of a paragraph on testis, apoptosis and oxidative stress may useful for the section.

Response: Thanks. Relevant information on codeine and testosterone has been added.

In Material and methods

- Most of method sections such as ‘Estimation of testicular enzymes’, ‘Estimation of testicular total protein and glucose’, ‘Estimation of serum and testicular levels of reproductive hormones’ and Estimation of testicular oxidative markers, anti-oxidants, 8OHdG and caspase 3 are too long, although ‘Histology’ section is too short. Please revise the sections.

Response: Thanks. These sections actually seem quite lengthy because a lot of parameters were assayed. This would be a useful reference for other researchers to aid their methodology.

- Company names and details of the commercial kits were detailed added to the manuscript, but there is no model, company, city and country details of the devices such as ELISA and spectrophotometer in the section. They should be added to the section.

Response: Thanks. These have been included

- In ‘8OHdG and caspase three levels were determined as described for AGE using ELISA kits

(Elabscience Biotechnology Co., Ltd, USA) with product numbers E-EL-0028 and E-EL’. The section is too short. Please use ‘caspase 3’ instead of ‘caspase three’ in the section.

Response: Thanks. ‘Caspase three’ has been replaced with ‘caspase 3’, and details of 8OHdG and caspase 3 assay has been added.

In results

- In ‘The decline in the relative testicular weight for codeine-treated rabbits at 4mg/kg and

10mg/kg body weight was 50% and 200% respectively below the control rabbits’. The sentence is not correct and it is conflicting with the results of Table 1. There are no 50% or 200% changes in the table.

Response: Thanks. This has been re-calculated and corrected

- In ‘Effect of chronic codeine use on testicular enzymes’. Please delete one of the double parenthesis in the ‘(GGT))’. SD of LDH in the low codeine group are is too high and I am waiting no difference between control and low codeine group according to the Figure 1 results.

Response: Thanks. This has been corrected.

There is no difference between the LDH in the low and high codeine groups. However, there is a statistical difference between the control and low codeine group, as well as between the control and high codeine group. This is evident by the error bars.

- For Figures 2 and 3 results, please add p values instead of the percentage changes.

Response: Thanks. The p values have been added

In discussion

- Most of abbreviations such as reduced glutathione (GSH), nitric oxide (NO) and cysteine-aspartic proteases (caspase) 3. were repeated here, although they we abbreviated in the introduction and material-method sections.

Response: Thanks. This has been modified.

- Discussion section is too short and it lacks focus. The author should concentrate on interpretation of their findings and their relevance to the field of study. The discussion should be used for the interpretation of data and for pointing out the significance of the findings.

Response: Thanks. This has been modified.

In Figures

- In figure 2. Please use antioxidants instead of anti-oxidants.

Response: Thanks. This has been corrected 

- In figure 8. There is no unit in the figure.

Response: Thanks. The unit has been added.

- In figure 9. Caspase 3 activity should be expressed as mg protein instead of ml.

Response: Thanks. The unit used is as stated in the manufacturer’s guideline.

- Superscripts such as a, b and c are not clear in the figures. For example, In figures 8 and 9, high dose codeine groups should contain b and c superscripts instead of only c.

Response: Thanks. When using such superscripts, each bar carries JUST ONE superscript, and the interpretation is stated below the figure. ‘Bars of the same parameter carrying different alphabets (a,b,c) are statistically different’ If bar one carries ‘a’, bar 2 carries ‘b’ and bar 3 carries ‘c’, it means that all the group are statistically different from one another. This is as seen in Figures 8 and 9.

If bar one carries ‘a’, bar 2 carries ‘b’ and bar 3 carries ‘b’, it means that groups 2 and 3 are similar but different from group 1. This is as seen in Figure 1. 

If If bar one carries ‘a’, bar 2 carries ‘a’ and bar 3 carries ‘a’, it means that all the groups are similar. This is as seen in Figure 2. 

It is a standard way of showing/representing significance .

- Graphical abstract is containing general and brief information. Please extend the graphics. For example, H2O2 is converted to water by GPx and the information should be present in the graphic.

 Response: Thanks. The graphical abstract summarizes JUST the findings of the present study. Adding details would make it cumbersome and would no longer be a graphical abstract.

Thanks.

With regards,

Ajayi AF, PhD

---

## [Editor Report · Decision Letter 2]

18 Dec 2019

PONE-D-19-27553R2

Testicular toxicity following chronic codeine administration is via oxidative DNA damage and up-regulation of TNF-α/NO and caspase 3

PLOS ONE

Dear Dr Ajayi,

Thank you for submitting your manuscript to PLOS ONE. After careful consideration, we feel that it has merit but does not fully meet PLOS ONE’s publication criteria as it currently stands. Therefore, we invite you to submit a revised version of the manuscript that addresses the points raised during the review process.

The following comments raised by the academic editor on the original manuscript should be addressed.

Additional Editor Comments (if provided):

The testicular toxicity due to drug abuse is well reported in many studies. However, the effect of codeine on testicular toxicity is not well understood and this manuscript provides basic on this aspect.

Additional experiments that provides an in depth analyses on the molecular and cellular mechanisms of codeine toxicity will add more strength to the study.

Statistical analyses are confusing and need to be carefully carried out. In many instances, there appears to be no significance, but it is indicated as significant.

Fructose is the main source of energy for spermatozoa. The rationale in measuring glucose is not provided. How does this fit?

Concrete evidence should be provided as to why testosterone decreases due to codeine treatment? Is it because of the altered levels of FSH / LH or any other cause?

Anti-oxidant enzyme activities should be presented as U/mg protein and not U/ml. Else it will be difficult to normalize the activities / levels between the samples.

Caspase-3 activity should be presented, not the levels (ng/ml).

We would appreciate receiving your revised manuscript by Feb 01 2020 11:59PM. To enhance the reproducibility of your results, we recommend that if applicable you deposit your laboratory protocols in protocols.io, where a protocol can be assigned its own identifier (DOI) such that it can be cited independently in the future. For instructions see: http://journals.plos.org/plosone/s/submission-guidelines#loc-laboratory-protocols

We look forward to receiving your revised manuscript.

Kind regards,

Suresh Yenugu

Academic Editor

PLOS ONE

Additional Editor Comments (if provided):

The following comments raised by the academic editor should be addressed in the revision

Additional Editor Comments (if provided):

The testicular toxicity due to drug abuse is well reported in many studies. However, the effect of codeine on testicular toxicity is not well understood and this manuscript provides basic on this aspect.

Additional experiments that provides an in depth analyses on the molecular and cellular mechanisms of codeine toxicity will add more strength to the study.

Statistical analyses are confusing and need to be carefully carried out. In many instances, there appears to be no significance, but it is indicated as significant.

Fructose is the main source of energy for spermatozoa. The rationale in measuring glucose is not provided. How does this fit?

Concrete evidence should be provided as to why testosterone decreases due to codeine treatment? Is it because of the altered levels of FSH / LH or any other cause?

Anti-oxidant enzyme activities should be presented as U/mg protein and not U/ml. Else it will be difficult to normalize the activities / levels between the samples.

Caspase-3 activity should be presented, not the levels (ng/ml).

---

## [Author Response · Author response to Decision Letter 2]

4 Feb 2020

Sir/ma,

REBUTTAL LETTER TO COMMENTS BY EDITOR AND REVIEWERS

The following comments raised by the academic editor on the original manuscript should be addressed.

Additional Editor Comments (if provided):

The testicular toxicity due to drug abuse is well reported in many studies. However, the effect of codeine on testicular toxicity is not well understood and this manuscript provides basic on this aspect.

Additional experiments that provides an in depth analyses on the molecular and cellular mechanisms of codeine toxicity will add more strength to the study.

Response: Thanks. More molecular and cellular mechanisms to buttress codeine-induced testicular toxicity has been added such as testicular DNA fragmentation and testicular TNF-α

Statistical analyses are confusing and need to be carefully carried out. In many instances, there appears to be no significance, but it is indicated as significant.

Response: Thanks. This has been clarified and well stated in the manuscript

Fructose is the main source of energy for spermatozoa. The rationale in measuring glucose is not provided. How does this fit?

Response: Thanks. Although fructose is hypothesized as the main energy source of sperm, lactate remains the preferred energy substrate which is primarily derived from glucose, and partly from fructose, via the glycolytic pathway. A detailed explanation with references has been included in the manuscript.

Concrete evidence should be provided as to why testosterone decreases due to codeine treatment? Is it because of the altered levels of FSH / LH or any other cause?

Response: Thanks. Concrete evidences have been clearly explained in the DISCUSSION section, and a summary in the CONCLUSION. 

Codeine depletes testicular glutathione, SOD, and catalase resulting in increased ROS production and inflammatory response which in turn leads to testicular lipid peroxidation, protein denaturation/inactivation, and testicular DNA damage, with consequent caspase-mediated testicular apoptosis and loss of testicular function.

Anti-oxidant enzyme activities should be presented as U/mg protein and not U/ml. Else it will be difficult to normalize the activities / levels between the samples.

Response: Thanks. This has been incorporated.

Caspase-3 activity should be presented, not the levels (ng/ml).

Response: Thanks. This has been incorporated.

Thanks for the comments. They added value to the manuscript.

---

## [Decision Letter · Decision Letter 3]

19 Feb 2020

PONE-D-19-27553R3

Testicular toxicity following chronic codeine administration is via oxidative DNA damage and up-regulation of NO/TNF-α and caspase 3 activities

PLOS ONE

Dear Dr Ajayi,

Thank you for submitting your manuscript to PLOS ONE. After careful consideration, we feel that it has merit but does not fully meet PLOS ONE’s publication criteria as it currently stands. Therefore, we invite you to submit a revised version of the manuscript that addresses the points raised during the review process.

We would appreciate receiving your revised manuscript by Apr 04 2020 11:59PM. To enhance the reproducibility of your results, we recommend that if applicable you deposit your laboratory protocols in protocols.io, where a protocol can be assigned its own identifier (DOI) such that it can be cited independently in the future. For instructions see: http://journals.plos.org/plosone/s/submission-guidelines#loc-laboratory-protocols

We look forward to receiving your revised manuscript.

Kind regards,

Suresh Yenugu

Academic Editor

PLOS ONE

Additional Editor Comments (if provided):

The concern on the way the statistical analyses are carried out is still there. For every table and figure the statement "Bars of the same parameter carrying different alphabets (a,b,c) are

statistically different" is given. This is a very vague statement. Clear indication should be given as to what a, b and c represent in all the tables and figures.

Reviewers' comments:

Reviewer's Responses to Questions

**Comments to the Author**

1. If the authors have adequately addressed your comments raised in a previous round of review and you feel that this manuscript is now acceptable for publication, you may indicate that here to bypass the “Comments to the Author” section, enter your conflict of interest statement in the “Confidential to Editor” section, and submit your "Accept" recommendation.

Reviewer #1: All comments have been addressed

2. Is the manuscript technically sound, and do the data support the conclusions?

Reviewer #1: Yes

3. Has the statistical analysis been performed appropriately and rigorously? 

Reviewer #1: Yes

4. Have the authors made all data underlying the findings in their manuscript fully available?

Reviewer #1: Yes

5. Is the manuscript presented in an intelligible fashion and written in standard English?

Reviewer #1: Yes

6. Review Comments to the Author

Reviewer #1: The authors perfromed all required comments in the manuscript and I have no further comment. Present form of the manuscript could be acceptable to the journal.

7. PLOS authors have the option to publish the peer review history of their article (what does this mean?). If published, this will include your full peer review and any attached files.

Reviewer #1: Yes: Prof. Dr. Mustafa Nazıroğlu

---

## [Author Response · Author response to Decision Letter 3]

21 Feb 2020

Thanks for the immense contribution to the work through your thorough review.

---

## [Editor Report · Decision Letter 4]

26 Feb 2020

Testicular toxicity following chronic codeine administration is via oxidative DNA damage and up-regulation of NO/TNF-α and caspase 3 activities

PONE-D-19-27553R4

Dear Dr. Ajayi,

We are pleased to inform you that your manuscript has been judged scientifically suitable for publication and will be formally accepted for publication once it complies with all outstanding technical requirements.

With kind regards,

Suresh Yenugu

Academic Editor

PLOS ONE
---

## [Editor Report · Acceptance letter]

2 Mar 2020

PONE-D-19-27553R4 

Testicular toxicity following chronic codeine administration is via oxidative DNA damage and up-regulation of NO/TNF-α and caspase 3 activities 

Dear Dr. Ajayi:

I am pleased to inform you that your manuscript has been deemed suitable for publication in PLOS ONE. Congratulations! Your manuscript is now with our production department. 

With kind regards,

on behalf of

Dr. Suresh Yenugu 

Academic Editor

PLOS ONE